# IBGraphRAG: Semantic Consistency and Information Bottleneck for Better Medical Knowledge Graph Retrieval

## Abstract

Large Language Models (LLMs) have achieved remarkable progress in recent years, but they still struggle to generate reliable and precise responses in domains such as medicine that rely heavily on specialized knowledge. Retrieval-Augmented Generation (RAG) offers a scalable solution for integrating external knowledge into LLMs without additional training, and while it performs well in general domains, its effectiveness is limited in medical settings that demand high terminological precision and factual consistency. To address this, we propose a medical-domain-oriented RAG framework, **IBGraphRAG**, which integrates medical knowledge graphs with two key technical innovations: (1) **Medical Semantic Consistency Alignment**, which improves entity recognition and linking by enforcing semantic consistency with a structured medical knowledge base; and (2) **Information Bottleneck-based Reasoning Path**, which prioritizes retaining highly relevant contextual information during knowledge graph retrieval while avoiding irrelevant or superficial paths. Experimental results show that IB-GraphRAG achieves state-of-the-art performance on multiple medical question-answering benchmark datasets, effectively improves the specialization and accuracy of the retriever in selecting reasoning paths over the knowledge graph, helping the LLM better identify relevant knowledge for reasoning.

## 1 Introduction

In recent years, large language models (LLMs) OpenAI (2023); Touvron et al. (2023); Team et al. (2023) have advanced rapidly, transforming natural language processing and powering many AI applications. However, they still face significant challenges in specialized domains like medicine Hadi et al. (2024); Williams et al. (2024); Xie et al. (2024). Medical knowledge is extensive and domain-specific, exceeding the context limits of current LLMs. While supervised fine-tuning can help, it is often costly or infeasible due to closed-source restrictions. Moreover, medicine requires precise terminology and strict factual accuracy (e.g., symptoms, drug side effects) that cannot be approximated. Model outputs must avoid distortion or fabrication, which is hard for non-experts to verify. Consequently, medical applications demand that LLMs perform complex reasoning over large external datasets and produce accurate, reliable, and verifiable responses.

To address these limitations, Retrieval-augmented Generation (RAG) Lewis et al. (2021) was introduced as a technique that allows LLMs to answer user queries using specific private datasets without requiring additional training. However, RAG has limited capacity for synthesizing new insights and performs poorly in tasks that require a holistic understanding of extensive documents. To overcome these shortcomings, GraphRAG Hu et al. (2024) was recently proposed. GraphRAG leverages LLMs to construct knowledge graphs from raw documents and enhances response generation by retrieving information from these graphs. By establishing clear conceptual relationships across data, GraphRAG demonstrates significant advantages over traditional RAG, particularly in tasks involving complex reasoning Hu et al. (2024). Recent work, such as SubgraphRAG Li et al. (2024a), has further improved upon this foundation by addressing the computational efficiency challenges faced by KG-based RAG methods. Although many KG-based RAG approaches have emerged Pan et al. (2024); Peng et al. (2024); Edge et al. (2024b), most of them are designed for general-purpose scenarios. Effectively applying these methods to the medical domain still presents challenges.

When applying KG-based methods to the medical domain, a major challenge lies in accurately identifying and aligning medical entities within user queries. Compared to general domains, medical terminology is highly specialized and fine-grained—different terms may refer to entirely distinct diseases, while the same concept may be expressed differently across regions or subfields. To address this issue, we propose **Medical Semantic Consistency Alignment** to enhance the accuracy of entity recognition in medical queries. This approach utilizes a general LLM to extract initial candidate entities, and then performs semantic consistency checks to align them with concepts in a structured medical knowledge base. Entities that semantically deviate from the query context are filtered out to avoid introducing irrelevant or misleading information. This ensures the knowledge graph retrieval is grounded in correct and contextually appropriate medical entities.

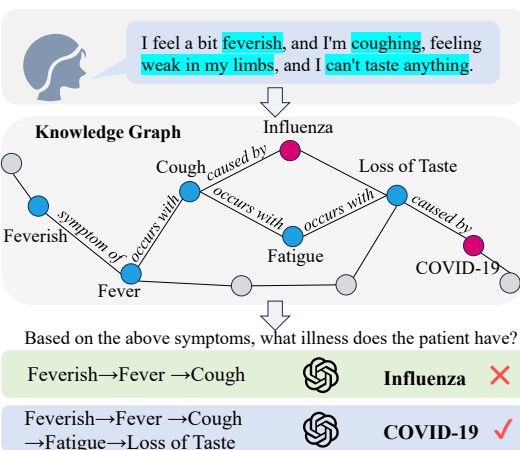

Figure 1: When a patient describes their symptoms, a misdiagnosis may occur if the RAG model fails to capture correct and sufficient information from the knowledge graph.

Another important challenge in the medical domain is the optimization of path selection strategies in knowledge graph traversal. Existing methods, such as SubgraphRAGLi et al. (2024a), typically adopt shortest-path heuristics between query and answer entities to improve retrieval efficiency. However, in medical knowledge graphs, such a heuristic path is not always the most informative or diagnostically meaningful. Medical entities often exhibit high semantic overlap,as shown in Figure 1, COVID-19 and influenza share similar symptoms such as fever and cough. Relying solely on the heuristic path may result in shallow retrieval with insufficient contextual support, which can lead to misinterpretation or vague answers. To address this issue, we propose **Information Bottleneck-based Reasoning Path**, which encourages the retriever to select paths that retain contextually relevant information while filtering out irrelevant content. Rather than prioritizing path brevity, our method focuses on retrieving paths that are more relevant to the question, thereby enhancing the model's ability to reason accurately and generate reliable responses in complex clinical scenarios.

Our main contributions can be summarized as follows:

- We systematically identify challenges in existing KG-based RAG methods when applied to specialized domains such as medicine, highlighting limitations in entity alignment and path selection.
- We propose two core techniques: **Medical Semantic Consistency Alignment** and **Information Bottleneck-based Reasoning Path**, improving the accuracy KG retrieval in medical settings.
- Extensive experiments on multiple medical datasets demonstrate that our RAG approach achieves significant improvements, outperforming a range of strong baseline models.

## 2 RELATED WORK

### 2.1 MEDICAL-DOMAIN LARGE LANGUAGE MODELS.

With the advancement of LLMs, several models specialized for the medical domain have emerged, such as BioGPT Luo et al. (2022), PMC-LLaMA Wu et al. (2023), BioMedLM Bolton et al. (2022), and Med-PaLM 2 Singhal et al. (2023). These models are fine-tuned using high-quality medical corpora and substantial computational resources provided by large organizations to enhance capabilities in medical reasoning and question answering. However, fine-tuning is costly and access is limited, making wide deployment challenging. Recent research has therefore focused on more cost-effective alternatives, particularly prompt engineering, which activates domain-specific knowledge without modifying model parameters Saab et al. (2024); Wang et al. (2023); Savage et al. (2024). In contrast, retrieval-augmented generation (RAG) has been widely studied in general domains but remains relatively underexplored in medical applications Miao et al. (2024); Xiong et al. (2024);

Long et al. (2024). Existing RAG methods perform well in open-domain tasks but lack fine-grained customization for clinical scenarios, resulting in insufficient terminological precision and factual consistency. IBGraphRAG addresses these gaps by explicitly aligning query entities with structured medical knowledge through *Medical Semantic Consistency Alignment (MSCA)*, improving both accuracy and robustness in complex clinical reasoning.

## 2.2 RETRIEVAL-AUGMENTED GENERATION.

RAG Lewis et al. (2021) enables models to use external datasets without additional fine-tuning, improving response accuracy and reducing hallucinations Guu et al. (2020). It has shown strong results across various tasks, including citation-supported generation Gao et al. (2023b); Slobodkin et al. (2024); Qi et al. (2024); Nakano et al. (2021); Bohnet et al. (2022); Gao et al. (2023a;c); Schimanski et al. (2024); Zhang et al. (2024b). GraphRAG Hu et al. (2024) enhances reasoning by constructing knowledge graphs, and SubgraphRAG Li et al. (2024a) builds upon this by optimizing subgraph retrieval. However, these methods typically rely on heuristic retrieval strategies that do not explicitly prioritize medically relevant reasoning paths, which limits their ability to surface diagnostically informative content and ensure factual accuracy. IBGraphRAG introduces an *Information Bottleneck-based Reasoning Path (IBRP)*, which selects clinically relevant reasoning paths while filtering out noisy or redundant links, thereby enhancing interpretability and supporting high-stakes medical decision-making.

## 3 METHODS

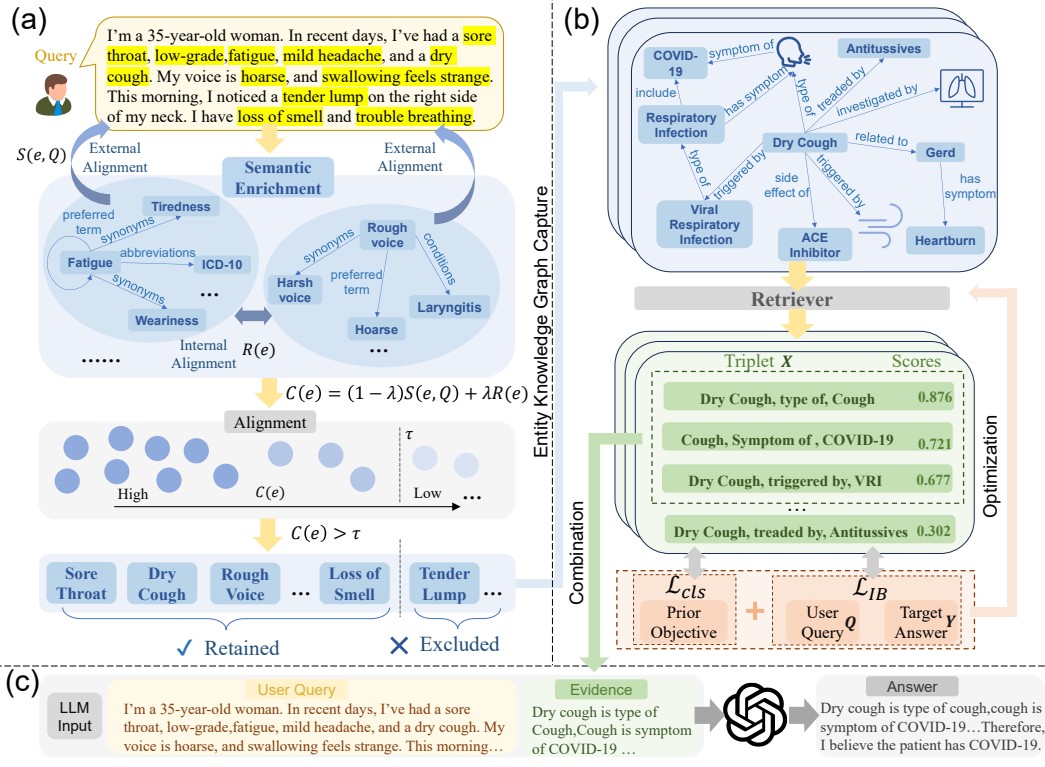

Figure 2: Our IBGraphRAG framework, (a) illustrates the Medical Semantic Consistency Alignment module, which performs entity extraction and alignment based on the user input. (b) shows the Information Bottleneck-based Reasoning Path retrieval module, which extracts evidence to support reasoning. (c) presents a simplified representation of LLM-based reasoning.

Our proposed framework adopts a *retrieval-and-reasoning pipeline* for KG-based RAG, as shown in Figure 2. Specifically, given a query $Q$, we first leverage a domain-specific medical LLM to

preliminarily extract medical entities from the query $A$, and then apply Medical Semantic Consistency Alignment (MSCA) to filter out entities that deviate from the intended medical semantics.and extract a subgraph $\mathcal{G}_q \subset \mathcal{G}$ from the knowledge graph that is relevant to these entities. Next, we employ the optimized retrieval method to obtain the Information Bottleneck-based Reasoning Path (IBRP) and retrieve knowledge most relevant to the query $Q$. Finally, the LLM generates responses based on the extracted relevant knowledge.

## 3.1 MEDICAL SEMANTIC CONSISTENCY ALIGNMENT

**Entities Extraction.** To train the retriever and enable knowledge graph querying, we first extract relevant entities. Given a question $Q$ and its answer $A$, the LLM identifies entities in $Q$ and $A$, yielding the sets $\{e_i^Q\}_{i \in [n]}$ and $\{e_j^A\}_{j \in [m]}$, where $n$ and $m$ are the numbers of entities in $Q$ and $A$, respectively. Using these, we query the knowledge graph $G$ to retrieve related entities. To avoid missing key entities, we adopt a multi-step strategy: a domain-specific medical modelZhang et al. (2019) encodes each entity $e \in E$ and computes its similarity to node embeddings in $G$, retaining the top-$K$ similar entities as a candidate set $S$. If relevant entities remain insufficient, in our experiments, we set 5 as the threshold for determining insufficient performance, the LLM further analyzes the QA context and entity names to refine selections. This process yields the mapped entity sets $\{\tilde{e}_i^Q\}_{i \in [n]}$ and $\{\tilde{e}_j^A\}_{j \in [m]}$ from $G$.

**Alignment.** After obtaining the initial entity list, it is necessary to ensure that the extracted entities do not exhibit significant semantic deviation and are medically relevant to the query $Q$. To achieve this, we leverage domain knowledge from the Unified Medical Language System (UMLS) Bodenreider (2004), enriching entity representations by incorporating synonyms, abbreviations, and semantically related terms. This process enhances the medical informativeness of the candidate entities, enabling more accurate filtering and improving the robustness of entity alignment.

Specifically, for each candidate entity $e \in \{e_i^Q\} \cup \{e_j^A\}$, we first retrieve its definition or semantic variants and synonyms from UMLS for semantic expansion. The enriched entity is then encoded into a dense vector representation $\mathbf{v}_e$ using a pre-trained biomedical embedding model, BioWordVec Zhang et al. (2019). Similarly, to maintain consistency in the embedding space, the query $Q$ is encoded into a vector $\mathbf{v}_Q$ within the same semantic space. We quantify the semantic similarity between each enriched entity and the query context using cosine similarity, defined as $S(e, Q) = \frac{\mathbf{v}_e \cdot \mathbf{v}_Q}{\|\mathbf{v}_e\| \|\mathbf{v}_Q\|}$. To promote semantic coherence among selected entities, we further introduce a regularization term $R(e)$ that captures the average pairwise similarity between $e$ and all other entities in the candidate set $S$, computed as $R(e) = \frac{1}{|S|-1} \sum_{e' \in S, e' \neq e} \frac{\mathbf{v}_e \cdot \mathbf{v}_{e'}}{\|\mathbf{v}_e\| \|\mathbf{v}_{e'}\|}$. The final alignment score $C(e)$ combines the local relevance of $e$ to the query and its global consistency within $S$, and is given by $C(e) = (1 - \lambda)S(e, Q) + \lambda R(e)$, where $\lambda \in [0, 1]$ is a tunable parameter balancing query-specific relevance and intra-set semantic cohesion. Here, we set $\lambda = 0.4$ to balance local semantic relevance and global coherence within the candidate entity set.

Entities with alignment scores $C(e)$ exceeding a threshold $\tau$ (set to $\tau = 0.6$) are retained to form the refined candidate set $S'$. These refined entities are then mapped to nodes in the knowledge graph $G$, yielding the final aligned entity sets $\{\hat{e}_i^Q\}_{i \in [n]}$ and $\{\hat{e}_j^A\}_{j \in [m]}$. This alignment strategy, reinforced by UMLS-based enrichment, ensures that only entities semantically aligned with the user query and contextually coherent within the medical ontology are utilized in downstream tasks.

## 3.2 INFORMATION BOTTLENECK-BASED REASONING PATH

**Information Bottleneck.** The information bottleneck (IB) principle Tishby et al. (1999) has been a great concept in finding a compression $\tilde{X}$ for signals $X$ that preserves the maximum information relevant to signals $Y$. Given a joint probability distribution $p(X, Y)$ between a random variable $X$ and an observed relevant variable $Y$ (with support sets $x \in \mathcal{X}$ and $y \in \mathcal{Y}$), the amount of information about $Y$ in compressed representation $\tilde{X}$ ($\tilde{x} \in \tilde{\mathcal{X}}$) is given by mutual information:

$$I(\tilde{X}; Y) = \int_{\tilde{\mathcal{X}}} \int_{\mathcal{Y}} p(\tilde{x}, y) \log \frac{p(\tilde{x}, y)}{p(\tilde{x})p(y)} \mathrm{d}\tilde{x}\mathrm{d}y, \tag{1}$$

where $I(\tilde{X}; Y) \leq I(X; Y)$ since compressed data can not convey more information than original ones. The information bottleneck is obtained through

$$\max \mathcal{L}_{\text{IB}} = I(\tilde{X}; Y) - \beta I(\tilde{X}; X), \tag{2}$$

where $\beta$ is the Lagrange multiplier for the trade-off between preserving meaningful information and compressing at various resolutions. However, the information bottleneck principle itself only enables compression and cannot transmit more information than the original data. To address this, we use it to guide reasoning path selection in the knowledge graph, aiming to identify paths most relevant to the target $Y$ while preserving the critical information from the signal $X$. Specifically, we construct a query graph based on the knowledge graph and optimize the selection of triples within it using the information bottleneck principle.

**Query Graph Construction.** In order to integrate the knowledge graph with known entities and perform reasoning path selection based on the information bottleneck principle, we first need to construct a representation that transforms the graph structure of the knowledge graph into a form interpretable by the model. To achieve this, we adopt a graph neural network (GNN) that performs message passing over entity, relation, and question embeddings (Yasunaga et al., 2021; Kang et al., 2023; Mavromatis & Karypis, 2024; Liu et al., 2024). Inspired by the SubgraphRAG approach (Li et al., 2024a), we further incorporate distance modeling between entities and topic entities to address multi-hop question answering tasks.

Specifically, given the topic entities $\mathcal{T}_Q$, we initialize each entity $e$ with a one-hot encoding $\mathbf{s}_e^{(0)}$ that indicates whether $e \in \mathcal{T}_Q$. At each iteration $(l+1)$, the feature propagation updates are computed by averaging the representations of all predecessor entities $e'$ connected to $e$, i.e., $\mathbf{s}_e^{(l+1)} = \text{MEAN}\{\mathbf{s}_{e'}^{(l)} \mid (e', \cdot, e) \in \mathcal{G}\}$. To capture the directed nature of the graph $\mathcal{G}$, reverse propagation is also performed, where the updated representation is given by $\mathbf{s}_e^{(r,l+1)} = \text{MEAN}\{\mathbf{s}_{e'}^{(r,l)} \mid (e, \cdot, e') \in \mathcal{G}\}$, and the initialization satisfies $\mathbf{s}_e^{(r,0)} = \mathbf{s}_e^{(0)}$. After propagating features in both directions, all intermediate representations are concatenated to form the final encoding of each entity as $\mathbf{s}_e = [\mathbf{s}_e^{(0)} \| \mathbf{s}_e^{(1)} \| \cdots \| \mathbf{s}_e^{(r,1)} \| \cdots]$. Finally, for each triple $\tau$, we construct its representation $z_\tau(\mathcal{G}, Q)$ by concatenating the encodings of its head and tail entities, i.e., $z_\tau(\mathcal{G}, Q) = [\mathbf{s}_h \| \mathbf{s}_t]$.

**Retriever Objective Specification.** Once the query graph is constructed, we select reasoning paths by defining a concrete and continuous training objective for the retriever. A direct application of the information bottleneck (IB) principle often leads to disjointed paths that lack a coherent reasoning trajectory, as they are merely associated with individual entities. To address this, we use the shortest path between the question and answer entities as a supervisory signal while introducing an IB-based optimization step to refine path selection. For efficient retrieval, we precompute semantic embeddings of all entities and relations in the knowledge graph using pre-trained text encoders Zhang et al. (2019); Beltagy et al. (2019) and store them in a vector database. Given a query $Q$, we encode it into $z_q$, retrieve embeddings $z_h, z_r, z_t$ for relevant entities and relations, and use an MLP to score and rank candidate triples. The retriever, integrating this MLP, is then jointly optimized with the supervisory signal and the IB objective, with the total loss defined as:

$$\mathcal{L}_{\text{total}} = \lambda_1 \left( -\frac{1}{N} \sum_{i=1}^{N} \left[ z_\tau^i \log \hat{z}_\tau^i + (1 - z_\tau^i) \log(1 - \hat{z}_\tau^i) \right] \right) + \lambda_2 \mathcal{L}_{\text{IB}} \tag{3}$$

where $N$ is the total number of candidate triples, and $z_\tau^i$ indicates whether the $i$-th triple belongs to the target reasoning path, $\hat{z}_\tau^i$ denotes the prediction probability output by the retriever. $\mathcal{L}_{\text{IB}}$ will be detailed in the following sections.

**IB-Guided Target Path Optimization.** After obtaining the preliminary set of triple-based reasoning paths $X$ filtered by the retriever, we transform these triples into language representations and perform a secondary embedding using a pre-trained language model. This allows us to further optimize the representations with respect to the target output $Y$ and the query $Q$ under the information bottleneck framework.

The IB objective in Eq. 2 consists of two parts. We first examine the first term $I(\tilde{X}; Y)$ in Eq. 2. This term measures the relevance between the compressed retrieved triples $\tilde{X}$ and the target output

$Y$. Given a query $Q$, we expand $I(\tilde{X}; Y|Q)$ as:

$$I(\tilde{X}; Y|Q) = H(Y|Q) - H(Y|\tilde{X}, Q) = -\int p(y, q) \log p(y|q)\, \mathrm{d}y\, \mathrm{d}q - H(Y|\tilde{X}, Q) \tag{4}$$

where the first term $\mathbb{E}_{(q,y)}[\log p(y|q)]$ is treated as a constant. Therefore, we can simplify it as:

$$I(\tilde{X}; Y|Q) \simeq -H(Y|\tilde{X}, Q) =: \int p(y, \tilde{x}, q) \log p(y|\tilde{x}, q)\, \mathrm{d}y\, \mathrm{d}\tilde{x}\, \mathrm{d}q \tag{5}$$

That is, we can approximate $I(\tilde{X}; Y \mid Q)$ using $\mathbb{E}_{(q,\tilde{x},y)}[\log p(y \mid \tilde{x}, q)]$. Instead of explicitly modeling $p(y|\tilde{x}, q)$ with large generative language models, we propose a lightweight relevance approximation by adopting a pretrained language encoding model $f_\theta$ to embed the retrieved triples $\tilde{X}$ and the target outputs $Y$ into a shared semantic space. In this space, the mutual information approximation is defined by the cosine similarity between the embeddings, i.e., $\mathbb{E}_{(q,\tilde{x},y)}[\cos(f_\theta(\tilde{x}), f_\theta(y))]$, and the corresponding optimization objective is formulated as:

$$\mathcal{L}_{\mathrm{sim}} = -\mathbb{E}_{(q,\tilde{x},y)}[\cos(f_\theta(\tilde{x}), f_\theta(y))] \tag{6}$$

This objective encourages the retriever to select triple paths whose semantic representations are aligned with the target outputs, thereby preserving relevant information for downstream tasks.

The second term $I(\tilde{X}; X|Q)$ in Eq. 2 aims to minimize the retention of irrelevant information while preserving critical components from the query $Q$, thus enabling the model to focus on generating higher-quality content. To make this term tractable, we introduce a variational upper bound by approximating the intractable conditional probability $p(x|\tilde{x}, q)$ with a parametric variational decoder $r_\psi(x|\tilde{x}, q)$. Details of the parametric variational decoder $r_\psi(x|\tilde{x}, q)$ are provided in the Appendix A.2. The mutual information $I(\tilde{X}; X|Q)$ can be expressed as:

$$\begin{aligned}
I(\tilde{X}; X|Q) &= \mathbb{E}_q\left[\int p(\tilde{x}, x|q) \log \frac{p(\tilde{x}, x|q)}{p(\tilde{x}|q)p(x|q)} \mathrm{d}\tilde{x}\mathrm{d}x\right] \\
&\leq \mathbb{E}_q\left[\int p(x|q) \log \frac{\int p(x|\tilde{x}, q)p(\tilde{x}|x, q)\mathrm{d}\tilde{x}}{p(x|q)} \mathrm{d}x\right],
\end{aligned} \tag{7}$$

Based on Jensen's inequality, we derive an upper bound for this term. This enables us to reformulate $I(\tilde{X}; X, Q)$ as a Kullback–Leibler (KL) divergence between the original retrieval probability distribution $p(x|q)$ and the expected reconstruction probability $p(x|\tilde{x}, q)$, where $\tilde{x}$ is integrated over the representation space defined by the noise filter $p(\tilde{x}|x, q)$. This formulation captures the extent to which the compressed representation $\tilde{x}$ preserves information relevant for reconstructing the associated triplet paths. Therefore, we have:

$$I(\tilde{X}; X \mid Q) \simeq \mathbb{E}_Q\left[D_{\mathrm{KL}}\left(p(x \mid Q) \,\|\, \mathbb{E}_{\tilde{X} \sim p(\tilde{X}|x, Q)}[p(x \mid \tilde{X}, Q)]\right)\right] \tag{8}$$

When queries and retrieved triple are jointly sampled from a training dataset $\{(q, x, y)\}$, the distribution $p(x|q)$ can be treated as a constant. Under this setting, the objective simplifies to:

$$\min I(\tilde{X}; X|Q) \simeq \min \mathbb{E}_{(q,x,\tilde{x})}[p(x|\tilde{x}, q)]. \tag{9}$$

Recent studies Federici et al. (2020); Kawaguchi et al. (2023); Fischer (2020) on the information bottleneck suggest replacing $I(X; \tilde{X})$ with $I(X; \tilde{X}|Y)$ to ensure that $I(X; \tilde{X})$ does not approach zero while retaining target-relevant information. Following these prior approaches, our final information bottleneck objective for path selection is defined as:

$$\begin{aligned}
\mathcal{L}_{\mathrm{IB}} &= I(\tilde{X}; X|Y; Q) - (\beta - 1)I(\tilde{X}; Y|Q) \\
&\simeq \mathbb{E}_{(q,x,\tilde{x},y)}[p(x|\tilde{x}, q, y)] - (\beta - 1)\mathbb{E}_{(q,\tilde{x},y)}[\log p(y|\tilde{x}, q)],
\end{aligned} \tag{10}$$

where the Lagrange multiplier satisfies $\beta - 1 > 0$.

### 3.3 REASONING

After fully training the retriever using the aforementioned methods, we leverage the retriever in combination with an LLM to perform reasoning. Specifically, for each user query $Q$, the retriever identifies entities within $Q$ and queries the knowledge graph for relevant information. The retrieved information is then processed into textual form and provided to the LLM together with the user's input for reasoning. Details of the LLM prompt design are provided in the Appendix A, where we also present a concrete example.

## 4 EXPERIMENTS

### 4.1 DATASETS

**RAG data.** Our experiment uses the Precision Medicine Knowledge Graph (PrimeKG) Chandak et al. (2023) as the external knowledge source. It integrates 20 high-quality biomedical resources to describe 17,080 diseases, containing 4,050,249 relationships covering ten major biological scales.

**Test Data.** Our test set are the test split of 9 multiple-choice biomedical datasets from the Multi-MedQA suite, 2 fact verification dataset about public health, i.e., FakeHealth Dai et al. (2020) and PubHealth Kotonya & Toni (2020). MultiMedQA includes MedQA Jin et al. (2021), MedMCQA Pal et al. (2022), PubMedQA Jin et al. (2019), and MMLU clinic topics Hendrycks et al. (2020). Detailed information about these datasets will be provided in the Appendix B.1 and Appendix B.2.

### 4.2 EXPERIMENT SETTING

**Baselines.** We tested six mainstream LLMs combined with different RAG methods, including Llama2 (13B, 70B), Llama3 (8B, 70B), Gemini-pro, and GPT-4. The Llama models were obtained from their official HuggingFace page. We used *gemini-1.0-pro* for Gemini-pro and *gpt-4-0613* for GPT-4. Our approach was compared with the standard RAG implemented by LangChain langchain (2024), GraphRAG Edge et al. (2024a) implemented by Microsoft Azure microsoft (2024), as well as MedGraphRAG Wu et al. (2024),SubGraphRAG Li et al. (2024a),KGARevion ?,KG-Rank Yang et al. (2024).

**Parameters and Model Configuration.** We use the vllm framework (Kwon et al., 2023) for efficient LLM inference. Since our approach employs a triple optimization strategy, we first score the triples and select the highest-scoring ones. In experiments, unless otherwise specified, all LLM reasoners use the top 100 retrieved triples; results using more triples are explicitly noted. For all LLM reasoners used in our experiments, both the temperature and random seed are set to 0 to ensure reproducibility. Some hyperparameter settings used in the paper have been provided in the Section 3.

We design empirical studies to systematically evaluate the effectiveness of IBGraphRAG in addressing the inherent challenges of KG-based RAG methods, covering both retrieval and reasoning aspects. Specifically, we aim to answer the following questions: **Q1**) Overall, compared to other RAG methods, can IBGraphRAG effectively improve accuracy on medical tasks? **Q2**) To meet the domain-specific requirements of fields like medicine, can IBGraphRAG precisely retrieve relevant information? **Q3**) Can the method in IBGraphRAG truly improve accuracy?

### 4.3 EXPERIMENTAL RESULTS AND ANALYSIS (**Q1** & **Q2**)

**Overall Performance (Q1).** Due to space constraints, we present results on a representative subset of datasets in the main text, with additional experimental results provided in the Appendix B.6. As illustrated in Table 1, our proposed method ( Orange ) consistently outperforms all baseline approaches across a diverse set of benchmarks. In particular, it demonstrates superior performance compared to models without retrieval-augmented generation (RAG) or employing standard RAG techniques ( Blue ), as well as more sophisticated retrieval-based variants such as GraphRAG and MedGraphRAG ( Red ), achieving state-of-the-art (SOTA) accuracy in nearly every evaluated scenario. These performance gains hold true across both multiple-choice medical question answering tasks and fact verification datasets, highlighting the robustness, stability, and broad generalizability

of our approach. Importantly, by applying our method, we significantly boost the reasoning capabilities of smaller-scale models (e.g., LLaMA-8B), enabling them to reach or even exceed the performance levels of much larger language models lacking RAG enhancements. This underscores not only the effectiveness but also the practical value of our framework, particularly in resource-constrained environments where deploying large-scale models may be prohibitive.

Table 1: Partial accuracy results of our method and all baseline methods. Values highlighted in  blue  indicate performance under no RAG, standard RAG, and other general RAG settings;  red  denotes the performance of GraphRAG and other RAG methods improved based on it;  orange  represents the results of our method. All best results are highlighted.

| Datasets | Multiple Choice QA | | | | | Fact Verification | |
|---|---|---|---|---|---|---|---|
| | MedQA | MedMCQA | PubMedQA | MMLU-Gene | MMLU-Clinic | Fake Health | Pub Health |
| Metrics | Acc. (std) | Acc. (std) | Acc. (std) | Acc. (std) | Acc. (std) | Acc. (std) | Acc. (std) |
| LLaMA2-13B | 0.426 (.009) | 0.375 (.008) | 0.678 (.010) | 0.661 (.008) | 0.637 (.015) | 0.536 (.007) | 0.495 (.009) |
| LLaMA2-13B (RAG) | 0.480 (.008) | 0.422 (.007) | 0.685 (.010) | 0.645 (.009) | 0.677 (.011) | 0.561 (.007) | 0.545 (.012) |
| LLaMA2-13B (KG-Rank) | 0.468 (.007) | 0.412 (.008) | 0.673 (.012) | 0.633 (.010) | 0.665 (.013) | 0.550 (.006) | 0.532 (.011) |
| LLaMA2-70B | 0.438 (.008) | 0.351 (.010) | 0.745 (.006) | 0.742 (.005) | 0.715 (.008) | 0.586 (.006) | 0.565 (.009) |
| LLaMA2-70B (RAG) | 0.565 (.006) | 0.500 (.007) | 0.749 (.008) | 0.760 (.006) | 0.735 (.011) | 0.641 (.012) | 0.629 (.013) |
| LLaMA2-70B (KG-Rank) | 0.551 (.007) | 0.488 (.005) | 0.735 (.010) | 0.748 (.007) | 0.722 (.012) | 0.629 (.010) | 0.617 (.011) |
| LLaMA3-8B | 0.600 (.008) | 0.575 (.007) | 0.755 (.012) | 0.828 (.018) | 0.750 (.011) | 0.510 (.007) | 0.534 (.008) |
| LLaMA3-8B (RAG) | 0.647 (.006) | 0.586 (.006) | 0.762 (.012) | 0.847 (.017) | 0.780 (.009) | 0.607 (.007) | 0.599 (.009) |
| LLaMA3-8B (KG-Rank) | 0.634 (.007) | 0.572 (.005) | 0.750 (.014) | 0.834 (.015) | 0.768 (.010) | 0.595 (.006) | 0.587 (.010) |
| LLaMA3-70B | 0.722 (.009) | 0.657 (.007) | 0.778 (.008) | 0.840 (.015) | 0.831 (.014) | 0.639 (.006) | 0.607 (.008) |
| LLaMA3-70B (RAG) | 0.825 (.005) | 0.727 (.005) | 0.809 (.010) | 0.869 (.006) | 0.872 (.008) | 0.762 (.006) | 0.722 (.007) |
| LLaMA3-70B (KG-Rank) | 0.811 (.008) | 0.715 (.004) | 0.796 (.012) | 0.857 (.005) | 0.860 (.009) | 0.751 (.008) | 0.709 (.006) |
| Gemini-pro | 0.592 (.008) | 0.551 (.007) | 0.701 (.010) | 0.755 (.008) | 0.770 (.014) | 0.604 (.006) | 0.639 (.009) |
| Gemini-pro (RAG) | 0.651 (.006) | 0.580 (.006) | 0.771 (.006) | 0.805 (.005) | 0.837 (.010) | 0.722 (.006) | 0.686 (.006) |
| Gemini-pro (KG-Rank) | 0.639 (.007) | 0.567 (.005) | 0.757 (.007) | 0.790 (.005) | 0.823 (.012) | 0.710 (.008) | 0.673 (.004) |
| GPT-4 | 0.785 (.008) | 0.730 (.006) | 0.755 (.015) | 0.902 (.007) | 0.865 (.008) | 0.715 (.009) | 0.711 (.012) |
| GPT-4 (RAG) | 0.884 (.005) | 0.769 (.005) | 0.778 (.004) | 0.930 (.006) | 0.931 (.010) | 0.788 (.004) | 0.776 (.007) |
| GPT-4 (KG-Rank) | 0.864 (.004) | 0.742 (.008) | 0.807 (.007) | 0.921 (.005) | 0.924 (.011) | 0.775 (.003) | 0.761 (.006) |
| KGARevion | 0.640 (.008) | 0.601 (.010) | 0.765 (.011) | 0.812 (.008) | 0.833 (.015) | 0.692 (.008) | 0.702 (.006) |
| LLaMA2-13B (GraphRAG) | 0.524 (.005) | 0.448 (.005) | 0.730 (.006) | 0.665 (.005) | 0.681 (.009) | 0.589 (.004) | 0.577 (.006) |
| LLaMA2-13B (MedGraphRAG) | 0.657 (.005) | 0.516 (.004) | 0.735 (.005) | 0.670 (.005) | 0.698 (.009) | 0.643 (.005) | 0.615 (.006) |
| LLaMA2-13B (SubGraphRAG) | 0.648 (.007) | 0.513 (.003) | 0.728 (.006) | 0.664 (.005) | 0.690 (.012) | 0.638 (.006) | 0.610 (.007) |
| LLaMA2-70B (GraphRAG) | 0.550 (.004) | 0.527 (.005) | 0.745 (.005) | 0.740 (.004) | 0.759 (.005) | 0.655 (.005) | 0.640 (.005) |
| LLaMA2-70B (MedGraphRAG) | 0.694 (.004) | 0.589 (.005) | 0.758 (.005) | 0.757 (.004) | 0.770 (.005) | 0.691 (.007) | 0.688 (.008) |
| LLaMA2-70B (SubGraphRAG) | 0.685 (.006) | 0.584 (.004) | 0.749 (.007) | 0.750 (.005) | 0.763 (.006) | 0.683 (.008) | 0.681 (.006) |
| LLaMA3-8B (GraphRAG) | 0.649 (.004) | 0.588 (.005) | 0.768 (.010) | 0.853 (.004) | 0.771 (.005) | 0.619 (.004) | 0.608 (.004) |
| LLaMA3-8B (MedGraphRAG) | 0.744 (.004) | 0.618 (.005) | 0.780 (.006) | 0.891 (.006) | 0.901 (.005) | 0.797 (.004) | 0.774 (.005) |
| LLaMA3-8B (SubGraphRAG) | 0.735 (.005) | 0.613 (.004) | 0.772 (.007) | 0.883 (.005) | 0.892 (.006) | 0.791 (.006) | 0.766 (.004) |
| LLaMA3-70B (GraphRAG) | 0.840 (.004) | 0.736 (.004) | 0.809 (.005) | 0.876 (.004) | 0.886 (.005) | 0.774 (.005) | 0.746 (.005) |
| LLaMA3-70B (MedGraphRAG) | 0.885 (.008) | 0.792 (.010) | 0.840 (.010) | 0.912 (.007) | 0.943 (.005) | 0.815 (.009) | 0.795 (.010) |
| LLaMA3-70B (SubGraphRAG) | 0.876 (.007) | 0.785 (.011) | 0.832 (.008) | 0.905 (.009) | 0.936 (.006) | 0.807 (.010) | 0.788 (.012) |
| Gemini-pro (GraphRAG) | 0.653 (.005) | 0.594 (.006) | 0.755 (.018) | 0.817 (.014) | 0.850 (.008) | 0.739 (.008) | 0.710 (.010) |
| Gemini-pro (MedGraphRAG) | 0.720 (.005) | 0.621 (.006) | 0.764 (.008) | 0.869 (.008) | 0.891 (.010) | 0.790 (.007) | 0.765 (.006) |
| Gemini-pro (SubGraphRAG) | 0.712 (.006) | 0.617 (.004) | 0.754 (.009) | 0.862 (.006) | 0.882 (.012) | 0.783 (.008) | 0.756 (.005) |
| GPT-4 (GraphRAG) | 0.890 (.005) | 0.774 (.004) | 0.782 (.004) | 0.922 (.005) | 0.941 (.010) | 0.785 (.005) | 0.780 (.005) |
| GPT-4 (MedGraphRAG) | 0.910 (.007) | 0.811 (.007) | 0.834 (.009) | 0.977 (.008) | 0.952 (.006) | 0.862 (.006) | 0.837 (.004) |
| GPT-4 (SubGraphRAG) | 0.902 (.008) | 0.804 (.009) | 0.829 (.006) | 0.964 (.010) | 0.939 (.007) | 0.848 (.005) | 0.832 (.004) |
| LLaMA2-13B (Ours) | 0.664 (.007) | 0.528 (.008) | 0.745 (.009) | 0.687 (.008) | 0.710 (.007) | 0.652 (.010) | 0.627 (.008) |
| LLaMA2-70B (Ours) | 0.705 (.006) | 0.601 (.009) | 0.771 (.010) | 0.769 (.008) | 0.780 (.009) | 0.702 (.007) | 0.695 (.010) |
| LLaMA3-8B (Ours) | 0.762 (.009) | 0.629 (.008) | 0.789 (.010) | 0.904 (.008) | 0.908 (.009) | 0.812 (.009) | 0.789 (.007) |
| LLaMA3-70B (Ours) | 0.897 (.010) | 0.804 (.009) | 0.852 (.010) | 0.922 (.010) | 0.951 (.008) | 0.830 (.009) | 0.802 (.007) |
| Gemini-pro (Ours) | 0.729 (.009) | 0.634 (.010) | 0.774 (.011) | 0.887 (.009) | 0.910 (.008) | 0.811 (.010) | 0.778 (.009) |
| GPT-4 (Ours) | 0.929 (.006) | 0.830 (.005) | 0.851 (.007) | 0.985 (.006) | 0.968 (.007) | 0.879 (.006) | 0.850 (.005) |

**Retrieval Analysis (Q2).** To verify that the performance improvements of our IBGraphRAG approach stem from effectively retrieving information relevant to medical queries—rather than merely introducing additional model parameters—we conduct a more detailed analysis of the knowledge graph content retrieved by our retriever.

To enable a more thorough and interpretable analysis of our method's behavior and effectiveness, we adopt an experimental setup distinct from our main experiments. While the primary setting retrieves the top 100 most relevant triples to provide the LLM with rich and diverse knowledge for answering queries, this larger retrieval size can make it difficult to clearly understand how the retrieval module contributes to performance improvements. Therefore, we reduce the retrieval size to 5 triples and retrain the retriever accordingly, allowing us to focus more precisely on the most essential and highly relevant information. By doing so, we can isolate the impact of key knowledge on the model's reasoning process. We further selected a representative subset of samples from the dataset to construct the training set. Additionally, we manually created a training example and used a

LLM to generate a corresponding knowledge graph, which was then integrated into the training data. For instance, the example query is: *"The patient has been coughing for three days, accompanied by fever and chest pain. Could these be symptoms of pneumonia?"* The answer is: *"Yes, these are typical symptoms of pneumonia. Pneumonia should be considered, and further examination is recommended."* Through visualization and qualitative analysis of this example, we aim to intuitively demonstrate the professionalism and precision of our approach within the medical domain. The related visualizations are presented in Figure 3.

By progressively visualizing the retrieval paths generated during training, we can observe the dynamic evolution of the retriever's behavior, as illustrated in Figure 3. The upper part of the figure shows the retrieval paths without IB optimization, while the lower part shows the paths under IB guidance. In the Target, the red points indicate the auxiliary optimization objectives, while the yellow points represent the selection points chosen by the model during the optimization process. In the early training stages, the retriever primarily learns to fit the predefined target paths, and this trend is consistent regardless of whether IB optimization is applied. However, once the retriever successfully learns these prior targets, our IB-based approach enables it to further refine the retrieval paths by selecting more informative and relevant knowledge. During testing, we find that because the IB training objective incorporates the entire query, the retriever learns to consider the full semantic context when determining optimal paths.

Under the objective without IB guidance, the retrieved path is: (cough, is a symptom of, upper respiratory infection), (upper respiratory infection, may be accompanied by, fever), (fever, may cause, elevated body temperature), (elevated body temperature, may lead to, chest discomfort), and (chest discomfort, suspected cause, pneumonia). This reasoning path does not accurately link to lung lesions and introduces ambiguous causal relationships such as "elevated body temperature" leading to "chest discomfort," which may result in misdiagnosis as common cold, anxiety, or viral infection. In contrast, our IB-guided reasoning path clearly links "cough" to "lower respiratory infection," avoiding confusion with common cold. It proceeds through (lower respiratory infection, may cause, alveolar congestion), (alveolar congestion, leads to, pulmonary consolidation), and (pulmonary consolidation, a typical pathological feature of, pneumonia), with the final connection (pneumonia, common symptom, chest pain) reinforcing the symptom's semantic closure. This path explicitly reflects the true pathological progression and provides a more precise and medically sound reasoning chain. This visualization not only demonstrates the effectiveness of our training strategy, but also offers valuable insights into how the retriever constructs meaningful knowledge paths—thereby improving both the interpretability and performance of downstream medical question-answering tasks.

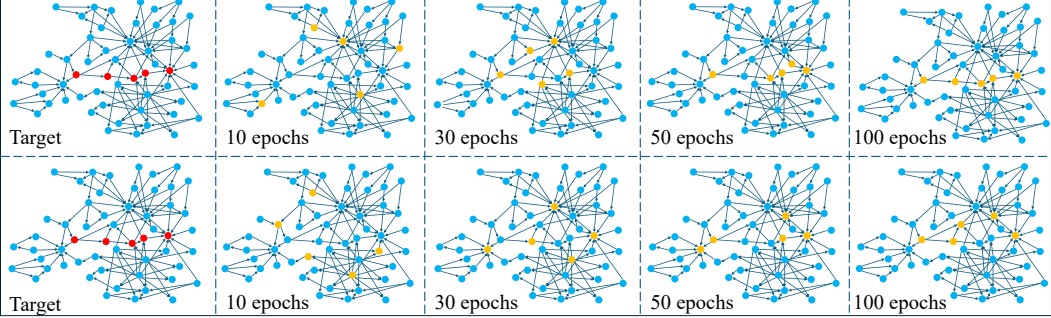

Figure 3: The visualization of the retriever's iterative convergence, with the upper row representing results without IB guidance and the lower row representing results with IB guidance.

## 4.4 ABLATION STUDY

To comprehensively evaluate the effectiveness of our proposed IBGraphRAG method and address **Q3**, we designed and conducted ablation experiments specifically targeting the retriever module. The overall ablation results are shown in the Table 2, clearly illustrating the contribution of each component to the model's overall performance. In addition, we further analyze the impact of the number of retrieved triples (i.e., retrieval scale) on model performance to gain a more detailed un-

derstanding of the relationship between retrieval depth and answer accuracy. The specific analysis results can be found in the Figure.

**Performance Analysis (Q3).** We conducted an overall analysis of the impact of each module on model performance, with the results shown in the Table 2. In our full system, we progressively removed the medical entity alignment module and the entropy-based optimization objective in the reasoning path. At the same time, to ensure a fair and controlled comparison, we retained key components such as the entity filtering strategy and the prior-guided objective for reasoning paths in the ablated versions. This carefully designed ablation setup allows us to more clearly identify and analyze the actual value and impact of each module in the context of medical QA tasks.

Table 2: Ablation study results with Accuracy (%) as the evaluation metric. Values in parentheses indicate the drop in accuracy compared to the full IBGraphRAG model.

| Model | Fake Health | Pub Health | MedQA | Med MCQA | Pub MedQA | MMLU Col-Med | MMLU Col-Bio | MMLU Pro-Med | MMLU Anatomy | MMLU Gene | MMLU Clinic |
|---|---|---|---|---|---|---|---|---|---|---|---|
| *Baselines with MSCA* | | | | | | | | | | | |
| Llama2-13B ↓ | $63.5_{(1.4)}$ | $60.6_{(2.1)}$ | $65.4_{(1.0)}$ | $50.7_{(2.1)}$ | $72.4_{(2.1)}$ | $68.3_{(1.8)}$ | $75.8_{(1.7)}$ | $66.5_{(2.4)}$ | $55.4_{(2.2)}$ | $66.6_{(2.5)}$ | $68.7_{(1.9)}$ |
| Llama2-70B ↓ | $68.6_{(1.6)}$ | $68.0_{(1.5)}$ | $68.7_{(1.8)}$ | $57.9_{(2.2)}$ | $75.2_{(1.9)}$ | $72.5_{(2.5)}$ | $87.9_{(1.7)}$ | $83.9_{(2.2)}$ | $68.1_{(2.5)}$ | $75.3_{(1.6)}$ | $76.4_{(1.6)}$ |
| Llama3-8B ↓ | $79.0_{(2.2)}$ | $77.3_{(1.6)}$ | $73.8_{(2.4)}$ | $60.9_{(2.0)}$ | $77.2_{(1.7)}$ | $88.6_{(1.6)}$ | $94.7_{(2.3)}$ | $90.9_{(2.2)}$ | $85.3_{(1.7)}$ | $88.6_{(1.8)}$ | $88.8_{(2.0)}$ |
| Llama3-70B ↓ | $80.2_{(2.8)}$ | $78.4_{(1.8)}$ | $87.3_{(2.4)}$ | $78.5_{(1.8)}$ | $82.7_{(2.5)}$ | $90.3_{(2.7)}$ | $95.4_{(2.4)}$ | $92.1_{(2.2)}$ | $88.6_{(2.8)}$ | $89.8_{(2.4)}$ | $92.9_{(2.2)}$ |
| Gemini-pro ↓ | $78.4_{(2.7)}$ | $75.8_{(2.0)}$ | $71.0_{(1.9)}$ | $61.1_{(2.3)}$ | $75.3_{(2.1)}$ | $85.7_{(2.3)}$ | $91.7_{(3.0)}$ | $88.8_{(2.1)}$ | $84.0_{(2.2)}$ | $86.0_{(2.8)}$ | $88.1_{(2.9)}$ |
| GPT-4 ↓ | $85.3_{(2.6)}$ | $82.7_{(2.3)}$ | $90.5_{(2.4)}$ | $80.7_{(2.3)}$ | $82.4_{(2.7)}$ | $90.8_{(2.2)}$ | $96.7_{(2.0)}$ | $94.8_{(2.1)}$ | $92.3_{(2.1)}$ | $97.7_{(0.8)}$ | $95.4_{(1.4)}$ |
| *Baselines without IBRP* | | | | | | | | | | | |
| Llama2-13B ↓ | $63.8_{(1.1)}$ | $60.9_{(1.8)}$ | $65.1_{(1.3)}$ | $51.1_{(1.7)}$ | $72.9_{(1.6)}$ | $68.1_{(2.0)}$ | $76.1_{(1.4)}$ | $66.9_{(2.0)}$ | $55.7_{(1.9)}$ | $67.0_{(2.1)}$ | $69.1_{(1.5)}$ |
| Llama2-70B ↓ | $68.9_{(1.3)}$ | $68.3_{(1.2)}$ | $68.9_{(1.6)}$ | $58.4_{(1.7)}$ | $75.7_{(1.4)}$ | $72.9_{(2.1)}$ | $88.2_{(1.4)}$ | $84.2_{(1.4)}$ | $68.6_{(2.0)}$ | $75.7_{(1.2)}$ | $76.9_{(1.1)}$ |
| Llama3-8B ↓ | $79.5_{(1.7)}$ | $77.5_{(1.4)}$ | $73.9_{(2.3)}$ | $61.3_{(1.6)}$ | $77.5_{(1.4)}$ | $88.9_{(1.3)}$ | $95.1_{(1.9)}$ | $91.3_{(1.8)}$ | $85.6_{(1.4)}$ | $88.9_{(1.5)}$ | $89.4_{(1.4)}$ |
| Llama3-70B ↓ | $80.9_{(2.1)}$ | $78.9_{(1.3)}$ | $88.1_{(1.6)}$ | $78.8_{(1.6)}$ | $83.5_{(1.7)}$ | $91.1_{(1.9)}$ | $96.2_{(1.6)}$ | $92.9_{(1.4)}$ | $89.5_{(1.9)}$ | $90.7_{(1.5)}$ | $93.8_{(1.3)}$ |
| Gemini-pro ↓ | $78.9_{(2.2)}$ | $76.1_{(1.7)}$ | $71.5_{(1.4)}$ | $61.7_{(1.7)}$ | $75.9_{(1.5)}$ | $86.0_{(2.0)}$ | $92.6_{(2.1)}$ | $89.4_{(1.5)}$ | $84.7_{(1.5)}$ | $86.8_{(1.9)}$ | $88.9_{(2.1)}$ |
| GPT-4 ↓ | $86.2_{(1.7)}$ | $83.1_{(1.9)}$ | $90.9_{(2.0)}$ | $81.2_{(1.8)}$ | $83.0_{(2.1)}$ | $91.2_{(1.8)}$ | $97.7_{(1.0)}$ | $95.5_{(1.4)}$ | $92.9_{(1.5)}$ | $97.2_{(1.3)}$ | $96.1_{(0.7)}$ |

It can be observed that removing any component from IBGraphRAG leads to a decline in performance. Notably, when the MSCA is removed, a significant drop in performance occurs on some models. For instance, on the MedQA dataset, Llama3-70B experiences a 2.6% decrease in accuracy. This aligns with our expectations: **if the question entities are incorrectly identified at the initial stage, causing semantic deviation, both subsequent retrieval and reasoning processes will be adversely affected.** Furthermore, removing the IBRP method and relying solely on the prior shortest path reasoning also results in performance degradation. This observation is consistent with the intuitive results, **more accurate, detailed, and relevant reasoning paths lead to noticeable improvements in LLM reasoning compared to simply using the shortest path.**

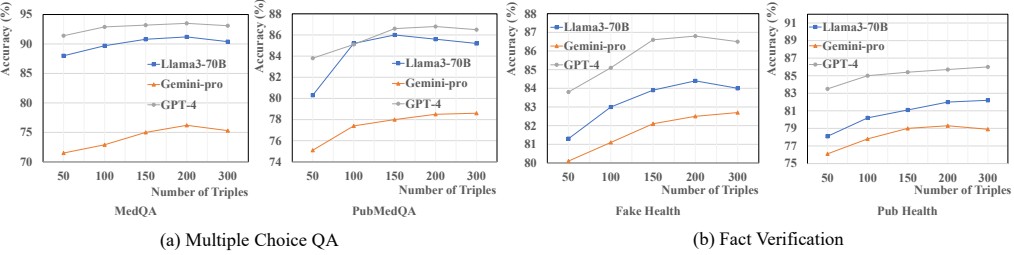

(a) Multiple Choice QA      (b) Fact Verification

Figure 4: The two figures on the left (a) show the results on multiple-choice QA with different retrieval scales, while the two figures on the right (b) present the results on fact verification under varying retrieval scales.

**Retrieval Size (Q3).** By varying the number of retrieved triples in the retriever, we further analyze the practical effectiveness of our proposed module in enhancing model performance. Specifically, we conduct experiments on four datasets, including the fact verification datasets *Fake Health* and *Pub Health*, and the multiple-choice QA datasets *MedQA* and *PubMedQA*, using retrieval sizes of 50, 100, 150, 200, and 300 triples. As shown in Figure 4, the reasoning performance of language models improves significantly as the number of retrieved triples increases from smaller scales. However, this performance gain gradually diminishes with further increases in retrieval size. On both multiple-choice and fact verification datasets, some LLMs even begin to exhibit performance degradation once the retrieval size exceeds a certain threshold. This suggests that an overly large

retrieval scope may undermine the effectiveness of the information bottleneck optimization objective, leading the retriever to include a substantial amount of irrelevant information, which ultimately harms the reasoning accuracy of the LLM. This trend indicates that our method does not depend on merely increasing the volume of retrieved information, but rather achieves performance improvements through more interpretable and contextually relevant path selection.

## 5 CONCLUSION

We propose IBGraphRAG, a knowledge graph-based retrieval-augmented generation framework. By integrating Medical Semantic Consistency Alignmen with Information Bottleneck-based Reasoning Path selection method, IBGraphRAG demonstrates superior performance in the medical domain. Extensive experimental results validate that our framework does not improve model reasoning simply by increasing the amount of retrieved information; rather, it effectively filters and selects relevant information, enabling the model to perform more rational and accurate reasoning. The results fully demonstrate the superiority of IBGraphRAG. The limitations of our work and future directions are provided in Appendix B.11.

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

APPENDIX

# A  IBGRAPHRAG

In this section, we provide additional details that could not be fully presented in the main text due to space limitations, in order to facilitate a more comprehensive understanding of the relevant aspects of our study.

## A.1  ENTITY RETRIEVAL

---

**Prompt for identifying entities**

**System:**
You are a highly knowledgeable and precise AI medical assistant specializing in extracting biomedical entities from text. Your goal is to identify and classify all relevant entities found within provided medical questions and answers. You must strictly adhere to the output format and entity type constraints described below.

---

**User:**
You are given two inputs: a medical question (Q) and its corresponding answer (A). Your task is to carefully extract all biomedical entities mentioned in both the question and the answer. Please ensure that the extraction is precise and comprehensive, including all relevant entities.
**Required Output Format** Return your results as a single JSON object containing two separate lists: one for entities extracted from the question (`"question_entities"`) and one for entities extracted from the answer (`"answer_entities"`).
Each entity must be represented as a JSON object with the following fields: `"id"`: a unique identifier string for the entity within its respective list.
`"type"`: the entity's type, which **MUST** be one of the following ten categories: 1.gene/protein, 2.drug, 3.effect/phenotype, 4.disease, 5.biological process, 6. molecular function, 7.cellular component, 8.exposure, 9.pathway, 10.anatomy.
`"name"`: the exact name of the entity as it appears in the text.
Please output the JSON exactly matching this structure, without any additional commentary or explanation.
**Example Output Structure**

```
{
  "question_entities": [
    {"id": "1", "type": "gene/protein", "name": "TP53"},
    {"id": "2", "type": "disease", "name": "lung cancer"}
  ],
  "answer_entities": [
    {"id": "1", "type": "drug", "name": "Aspirin"}
  ]
}
```

---

Figure 5: The prompt configuration for initially extracting entities from user questions and answers

In the main body of our work, we leverage LLMs to extract and retrieve relevant entities from both questions and answers, providing supervision signals for training the retriever. However, how to effectively apply LLMs for entity extraction remains a non-trivial challenge. Prior studies have shown that the performance of LLMs is highly sensitive to the design of prompts. Even minor modifications in prompt phrasing can lead to significant changes in output behavior Wei et al. (2022); Li et al. (2024b); Zhang et al. (2024c); Feng et al. (2024). Therefore, prompt design plays a crucial role in ensuring that LLMs can accurately and consistently identify entities within a given QA context Shin et al. (2020); Deng et al. (2022); Yao et al. (2024); Zhang et al. (2024a). This issue should also be carefully considered when leveraging LLMs to retrieve semantically similar entities from the query.

To support reproducibility and facilitate further research, we provide the exact prompt templates used in our experiments in this section. While we acknowledge the impact of prompt design on extraction performance, this aspect is not the central focus of our work, and we thus do not conduct dedicated ablation studies on prompt variation. The prompt configuration for initial entity extraction is shown in Figure 5. We adopt a strategy inspired by MedReason Wu et al. (2025), where we constrain the reasoning scope provided to the LLM. By explicitly defining the contextual range, typically limited to the user question and its corresponding answer, we guide the model to perform entity extraction more accurately within a medically relevant boundary. This targeted approach helps prevent the model from identifying irrelevant or non-biomedical entities, such as personal names or institution names, which often carry no meaningful value in the context of medical analysis.

---

**Prompt for identifying most related entities**

**System:**
You are a precise and context-aware biomedical assistant designed to help users perform entity-level reasoning tasks in medical question-answering scenarios. You must follow instructions strictly, return outputs only in the specified format, and always ensure that your responses are medically relevant, factual, and grounded in the provided information. Do not make assumptions beyond the given context.

---

**User:**
You are given the following inputs: a medical question (Q), its corresponding answer (A), a `query_entity` extracted from the question, and a list of similar entities related to the query entity.
Your task is to select **one** most relevant entity from the list of similar entities based on its contextual and semantic alignment with the question-answer pair. If none of the similar entities is appropriate in the given context, return NONE.

**Required Output Format:**
Return your answer as a single JSON object using the following structure.
Each selected entity must be represented as a JSON object with the following fields: Each selected entity must include the following three fields: `"name"`, which indicates the name of the selected entity from the similar entities list; `"id"`, an integer representing the index of the selected entity within the similar entities list (starting from 0); and `"reason"`, a brief justification (1–2 sentences) explaining why this entity was chosen based on the question, answer, and query entity.
If no appropriate entity is found in the list, return:

```
{
    "selected_entity": "NONE"
}
```

Otherwise, follow the format below precisely. Do not include any additional commentary or explanation.

**Example Output Structure**

```
{
    "selected_entity": {
        "name": "CYP2D6",
        "id": 3,
        "reason": "CYP2D6 is mentioned in the answer and is related to drug metabolism."
    }
}
```

Figure 6: The prompt configuration for extracting related entities from user questions and answers

When the initial entity extraction step does not yield enough entities to effectively support downstream retrieval tasks, we expand the candidate set by searching for semantically similar entities within the knowledge graph based on the initially extracted ones. Although this semantic similarity-based expansion helps to increase coverage, it often results in an excessively large number of entities, which can lead to an overwhelming retrieval scope and potential noise—outcomes that are undesirable. To address this, we apply a second-stage filtering using a language model to refine the expanded list by selecting only the most contextually relevant entities. This LLM-guided selection helps ensure that the final set of entities used for retrieval is both manageable in size and highly pertinent. The prompt used for this filtering process is detailed in Figure 6. Of course, in the subsequent medical entity alignment step, we perform more fine-grained and precise filtering to ensure accurate matching and high-quality alignment of entities. However, prior to that, we effectively prevent the introduction of excessive noise entities by designing the prompt as described, thereby controlling the scale of candidate entities from the source. This not only improves the efficiency of entity filtering but also reduces unnecessary aggregation interference during the subsequent alignment process, ensuring that the medical entity alignment operates on a clearer and cleaner set of entities, ultimately enhancing the overall accuracy and robustness of the system.

## A.2    IB-GUIDED TARGET PATH OPTIMIZATION

In Section 3, we have presented a comprehensive framework for optimizing knowledge graph path selection grounded in the information bottleneck principle. Nevertheless, certain intricate aspects and implementation details were not fully articulated in the main text. In this section, we aim to further elaborate on these components to provide a more thorough and nuanced understanding of the methodology as well as its inherent advantages.

As discussed previously, the direct computation of the conditional mutual information $I(\tilde{X}; X \mid Q)$ poses significant intractability challenges due to the complexity of the underlying distributions. To effectively address this, we introduce a parametric variational decoder, denoted as $r_\psi(x \mid \tilde{x}, q)$,

which serves to approximate the true conditional probability distribution $p(x \mid \tilde{x}, q)$. By doing so, we derive a tractable upper bound on the conditional mutual information, facilitating feasible optimization. Although the main text outlines the general framework of this approach, it omits a detailed account of the variational decoder's training procedure. Here, we provide a systematic and comprehensive description of the training methodology and architectural specifics of the variational decoder $r_\psi(x \mid \tilde{x}, q)$.

The training process of the parametric variational decoder relies on the existing dataset $\{(q, x)\}$, where $q$ denotes the input query and $x$ corresponds to the associated knowledge graph path. The optimized representation $\tilde{x}$ of the knowledge graph path is derived from preliminary filtering results obtained through the retriever module. This representation $\tilde{x}$, concatenated with the query $q$, serves as the input to the variational decoder. Architecturally, $r_\psi$ is formulated as a conditional generative model parameterized by $\psi$. This model processes the concatenated embeddings of $\tilde{x}$ and $q$ and outputs a probability distribution over the original target space of $x$. This design enables the decoder to model the conditional likelihood function $r_\psi(x \mid \tilde{x}, q)$ and thus reconstruct the original entity $x$ from the compressed representation and query context.

The objective function for training the variational decoder is formulated as maximizing the expected log-likelihood of reconstructing the original entity $x$ given $\tilde{x}$ and $q$, which can be expressed as $\max_\psi \mathbb{E}_{(q,x)}\mathbb{E}_{\tilde{x}\sim p(\tilde{x}|x,q)}\left[\log r_\psi(x \mid \tilde{x}, q)\right]$. This objective equivalently maximizes a variational lower bound on the otherwise intractable conditional mutual information $I(\tilde{X}; X \mid Q)$, thus encouraging the variational decoder to faithfully reconstruct and preserve the informative features of the original knowledge graph paths that are essential for downstream retrieval performance.

During the training phase, mini-batches of $(q, x)$ pairs are sampled from the dataset. For each pair, the compressed representation $\tilde{x}$ is sampled according to the known noise distribution $p(\tilde{x} \mid x, q)$. The parameters $\psi$ of the decoder are optimized by backpropagating the reconstruction loss using stochastic gradient-based optimizers, such as Adam. This training protocol ensures that the variational decoder progressively approximates the true conditional distribution $p(x \mid \tilde{x}, q)$, enabling efficient and tractable estimation of the upper bound on $I(\tilde{X}; X \mid Q)$ during the overall model optimization.

## A.3 REASONING

As previously discussed, the design of prompts plays a crucial role in influencing the reasoning performance and output quality of LLMs. In recent years, a variety of reasoning paradigms—most notably Chain-of-Thought (CoT) prompting—have emerged, underscoring the growing importance of prompt engineering. In this work, we further explore how structured knowledge in the form of knowledge graph triples can be leveraged to guide LLMs toward more interpretable and accurate reasoning trajectories. Specifically, we construct prompts that explicitly encode entity-relation-entity triples, encouraging the model to perform step-by-step inference along these semantic paths. This design enables the LLM to engage in structured CoT reasoning, grounded in the relational semantics of the input knowledge. As a result, our method not only enhances the model's reasoning accuracy, but also significantly improves interpretability by making intermediate reasoning steps more transparent and logically coherent. By incorporating structured knowledge into the prompting process, we provide a principled and effective approach to supporting complex query tasks through LLM-based reasoning. Our prompt setup is shown in Table 3.

Table 3: Prompt template used during reasoning.

---

**System**    Given the set of retrieved triples from the knowledge graph, generate an answer to the question. Your response should be a list of items, each beginning with "ans:".

**User**    Given the following structured knowledge in the form of triplets: $(e_a, r_{ab}, e_b)$, $(e_b, r_{bc}, e_c)$, $(e_c, r_{cd}, e_d)$, ... Question: ...

**Assistant**    To answer the question, we have to find .... From the triples we can see that .... Therefore, the answers are: ans: ... ans: ...

---

To facilitate readers' deeper understanding, we provide a more detailed example. A detailed example can be found in Figure 7. Here, we employ In-Context Learning (ICL) Mann et al. (2020) to provide examples that guide the LLM to generate answers that are more accurate and better aligned with our expectations. By leveraging ICL, the model's reasoning capabilities are enhanced, leading to improved relevance and consistency in the generated responses, thereby demonstrating superior performance in practical applications.

---

**Input Prompts for KGQA**

**System:**
Given the set of retrieved triples from the knowledge graph, generate an answer to the question. Your response should be a list of items, each beginning with "ans:".

---

**User:**   // ICL example
Given the following structured knowledge in the form of triplets:
(Aspirin, medical.drug.treats, Headache)
(Headache, medical.condition.symptom_of, Migraine)
(Migraine, medical.condition.caused_by, Cortical_Spreading_Depression)
(Cortical_Spreading_Depression, medical.condition.related_to, Neurological_Disorder)
(Migraine, medical.condition.treated_by, Triptans)
(Triptans, medical.drug.mechanism, Serotonin_Receptor_Agonist)
(Aspirin, medical.drug.side_effect, Stomach_Ulcer)
(Stomach_Ulcer, medical.condition.caused_by, NSAIDs)
(NSAIDs, medical.drug.class, Anti_Inflammatory)
(Migraine, medical.condition.symptom_of, Vascular_Disorder)
...
Question:
Which drug that acts as a serotonin receptor agonist is used to treat the neurological disorder related to cortical spreading depression causing migraine?

---

**Assistant:**   // ICL example
To answer the question, we first identify the neurological disorder related to cortical spreading depression.
From the triplets, cortical spreading depression is related to neurological disorder and causes migraine.
Next, we find the drug that treats migraine and acts as a serotonin receptor agonist.
The triplets indicate that Triptans treat migraine and their mechanism is serotonin receptor agonist.
Therefore, the drug acting as a serotonin receptor agonist used to treat the neurological disorder related to cortical spreading depression causing migraine is Triptans.
Thus, the formatted answer is:
ans: Triptans (serotonin receptor agonist treating migraine)

---

**User:**   // Sample to infer
Given the following structured knowledge in the form of triplets:
$(e_a, r_{ab}, e_b)$   // Retrieved triples
$(e_c, r_{cd}, e_d)$
. . .
Question:
what's . . . ?   // Q in datasets

Figure 7: Detailed prompt for LLM question-answering used in our experiments for all samples.

## B  EXPERIMENTS

### B.1  RAG DATA

In the main body of the paper, we have briefly introduced the overall structure and functional role of the knowledge graph utilized in this study. To facilitate a deeper understanding of this resource, this section provides a more systematic and detailed exposition of its construction methodology, semantic organization, and the distribution of entity and relation types. PrimeKG Chandak et al. (2023) is a large-scale heterogeneous biomedical knowledge graph specifically designed to support downstream reasoning tasks in the domain of precision medicine. PrimeKG integrates structured knowledge from over 20 curated biomedical databases, including UMLS, DrugBank, DisGeNET, OMIM, and Reactome, resulting in a unified graph comprising approximately 90,000 entities and over 1.3 million relational triples. The knowledge graph covers 17 distinct entity types (e.g., diseases, drugs, genes, anatomical structures) and 49 relation types (e.g., *treats*, *associated_with*, *encodes*), facilitating rich semantic connectivity. Each entity is accompanied by textual descriptions and type annotations, and many nodes are aligned with external biomedical vocabularies to ensure interoperability. Due to its comprehensive coverage, high-quality curation, and multi-relational structure, PrimeKG serves as a

robust foundation for biomedical knowledge representation and reasoning tasks such as multi-hop question answering, drug repurposing, and disease mechanism inference. The types and distribution of relations contained in the knowledge graph are summarized in Table 17. If we directly input the relation types here into the language model, we are concerned that, due to the nature of its training process, the model may not accurately capture the specific meanings of these relations. To address this issue, we provide detailed and explicit descriptions for each relation type, enabling the LLM to better understand their semantics and usage. For the specific descriptions, please refer to the Table 16.

## B.2 TEST DATA

We evaluate our system on a diverse suite of health-related benchmarks, categorized into two main groups: **MultiMedQA** and **Fact Verification Datasets**. The former includes multiple-choice question answering (MCQA) datasets drawn from standardized medical licensing exams, clinical case assessments, and biomedical literature, aiming to evaluate a model's ability in factual recall, diagnostic reasoning, and domain-specific understanding. The latter focuses on the verification of public health claims and health-related misinformation, requiring models to assess the truthfulness of real-world statements based on supporting evidence or expert justification.

**MultiMedQA.** The MultiMedQA benchmark comprises a diverse collection of biomedical multiple-choice question answering (MCQA) datasets, including both standalone resources and medical-related subsets of the MMLU benchmark. These datasets span a broad range of topics such as clinical medicine, biomedical research, human anatomy, and genetics. Each question consists of a single prompt followed by four answer choices, reflecting the format of real-world standardized medical assessments. Tables 4 and 5 summarize the datasets and provide representative examples.

| Dataset | Description |
|---|---|
| MedQA | Multiple-choice questions from the Chinese National Medical Licensing Examination, requiring expert-level clinical reasoning. |
| MedMCQA | A large-scale QA dataset based on Indian AIIMS and NEET PG medical entrance exams, covering multi-hop and multi-sentence reasoning. |
| PubMedQA | Questions and answer options automatically generated from PubMed article abstracts and manually verified, testing biomedical research comprehension. |
| MMLU–Col-Med | Medical knowledge questions at the college level from the MMLU benchmark, focused on general health sciences. |
| MMLU–Col-Bio | Undergraduate-level biology questions from the MMLU suite, covering molecular, cellular, and systemic biology. |
| MMLU–Pro-Med | Professional medical exam-style questions (e.g., USMLE), assessing clinical diagnostics and therapeutics. |
| MMLU–Anatomy | Questions focusing on human anatomical structures and spatial relationships, derived from medical school curricula. |
| MMLU–Gene | Assessment of foundational and applied genetics concepts, including heredity and molecular biology. |
| MMLU–Clinic | Clinical case–based questions simulating real-world diagnostic and treatment decisions. |

Table 4: Descriptions of the datasets included in the MultiMedQA benchmark.

**Fact Verification Datasets.** This group of datasets evaluates the model's ability to assess the factual accuracy of health-related claims, based on supporting or refuting evidence. The claims are labeled with fine-grained veracity judgments (e.g., True, False, Misleading) and are accompanied by expert-written or evidence-based justifications. Tables 6 and 7 summarize the dataset characteristics and provide representative examples.

## B.3 DATA INPUT

| Dataset | Question (with Options) |
|---|---|
| MedQA | Which of the following is the most appropriate management for a patient with acute ST-elevation myocardial infarction? 
 **A.** Oxygen therapy **B.** Aspirin **C.** Intravenous fluids **D.** Antihypertensives |
| MedMCQA | Which vitamin deficiency leads to night blindness? 
 **A.** Vitamin A **B.** Vitamin B1 **C.** Vitamin C **D.** Vitamin K |
| PubMedQA | Does zinc supplementation improve common cold outcomes? 
 **A.** Yes, significantly **B.** No **C.** Only in children **D.** Not enough evidence |
| MMLU–Col-Med | What is the function of the sinoatrial (SA) node in the heart? 
 **A.** Pumps blood **B.** Initiates heartbeat **C.** Filters blood **D.** Contracts ventricles |
| MMLU–Col-Bio | Which organelle is primarily responsible for energy production in a cell? 
 **A.** Nucleus **B.** Golgi apparatus **C.** Mitochondria **D.** Lysosome |
| MMLU–Pro-Med | A 25-year-old male presents with fatigue and sore throat. Monospot test is positive. Likely diagnosis? 
 **A.** Influenza **B.** Streptococcal pharyngitis **C.** Infectious mononucleosis **D.** COVID-19 |
| MMLU–Anatomy | The femoral artery passes through which anatomical structure? 
 **A.** Foramen magnum **B.** Inguinal canal **C.** Adductor canal **D.** Sciatic notch |
| MMLU–Gene | What base does adenine pair with in DNA? 
 **A.** Cytosine **B.** Thymine **C.** Guanine **D.** Uracil |
| MMLU–Clinic | A diabetic patient presents with confusion and fruity breath. What is the most likely diagnosis? 
 **A.** Hypoglycemia **B.** Hyperosmolar state **C.** Diabetic ketoacidosis **D.** Renal failure |

Table 5: Example questions from MultiMedQA datasets.

| Dataset | Description |
|---|---|
| FakeHealth | A benchmark for health misinformation detection. It contains health-related news headlines or stories annotated with binary factuality labels (True/Fake) and stance information (Support/Neutral/Oppose), simulating a social media–like environment. |
| PubHealth | A fact-checking dataset consisting of public health–related claims labeled with nuanced factuality categories (e.g., True, False, Misleading), along with manually curated expert justifications, designed to reflect real-world health misinformation scenarios. |

Table 6: Descriptions of the datasets used for health-related fact verification.

We provide a detailed description of the two tasks used in our evaluation—Medical Question Answering (MedQA) and Medical Fact Verification—and explicitly explain how the components of each task are mapped into the IB-guided target path optimization process.

**Medical Question Answering**

**Input:** A natural language medical question and multiple candidate answers.

**Output:** The correct answer option (e.g., disease, treatment, or clinical concept).

**Entity Extraction:** Symptom entities are extracted from the question, while target disease or treatment entities are extracted from candidate answers.

**Mapping to IB-guided Optimization:** Query entities are aligned to nodes in the medical knowledge graph, and answer entities serve as target nodes. The retriever constructs reasoning paths from

| Dataset | Claim, Label, and Justification |
|---------|--------------------------------|
| FakeHealth | Claim: Flu vaccine causes Alzheimer's disease. 
 **Label:** Fake 
 **Justification:** No biological evidence links flu shots to Alzheimer's. |
| FakeHealth | Claim: WHO approved a new malaria vaccine. 
 **Label:** True 
 **Justification:** Confirmed via WHO official announcement. |
| PubHealth | Claim: Vitamin C cures the common cold. 
 **Label:** False 
 **Justification:** Clinical studies show it may reduce duration but not cure the illness. |
| PubHealth | Claim: 5G radiation weakens the immune system. 
 **Label:** Misleading 
 **Justification:** No scientific basis; refuted by global health authorities. |

Table 7: Example claims from fact verification datasets.

query to answer entities, while the IB-guided optimization ensures that selected paths maximize relevance (e.g., symptom → disease) and filter out superficial or noisy links.

**Medical Fact Verification**

**Input:** A claim (e.g., "Vitamin C cures influenza") and supporting/rebuttal evidence documents.

**Output:** A binary label (Supported / Refuted).

**Entity Extraction:** Biomedical entities are extracted from the claim, and related entities are extracted from the evidence documents.

**Mapping to IB-guided Optimization:** Claim entities are treated as query nodes and evidence entities as candidate supporting nodes. IB-guided optimization selects reasoning paths that preserve medically relevant causal or evidential relations (e.g., "Vitamin C → clinical trial → no significant effect → influenza"), ensuring factual grounding.

**Training vs. Inference**

During training, explicit target entities are provided. At inference time, only entities from the question or claim are available. Through IB-guided optimization during training, the retriever learns to effectively identify medically relevant reasoning paths without requiring answer entities at inference.

B.4 BASELINE

**Llama2 (13B, 70B) Touvron et al. (2023)** Llama2, developed by Meta, is a series of open-source large language models that build upon the original LLaMA architecture. The 13B and 70B versions denote models with 13 billion and 70 billion parameters, respectively. These models are pretrained on massive internet text corpora, exhibiting strong language understanding and generation capabilities. Their open-source nature facilitates widespread adoption in both academic and industrial applications, making them effective backbones for Retrieval-Augmented Generation (RAG) systems, especially when integrated with retrieval mechanisms to enhance knowledge grounding.

**Llama3 (8B, 70B) Dubey et al. (2024)** Llama3 represents the next generation of Meta's large language models, improving upon Llama2 with enhanced architecture and training techniques. The 8B and 70B parameter models cater to different computational resource constraints and performance needs. Llama3 offers better inference speed, context comprehension, and multitask capabilities. It also incorporates stricter data curation and safety measures, resulting in more accurate and ethically aligned outputs, which is particularly valuable in specialized domains like healthcare and law within RAG frameworks.

**Gemini-pro (gemini-1.0-pro) Team et al. (2023)** Gemini-pro is part of Google's advanced large language model lineup, emphasizing multimodal fusion and large-scale knowledge understanding. The 1.0-pro version signifies a professional-grade model with efficient reasoning and superior se-

mantic representation capabilities. Gemini-pro excels in integrating text and image modalities, making it well-suited for tasks requiring cross-modal knowledge retrieval and generation. When combined with RAG, it effectively consolidates retrieved information to produce highly relevant and accurate responses, especially for complex reasoning and long-form generation.

**GPT-4 (gpt-4-0613) Achiam et al. (2023)** GPT-4 is OpenAI's state-of-the-art flagship language model, renowned for its exceptional language understanding and generation skills. The 0613 variant is a stable API release optimized for diverse tasks. GPT-4 demonstrates superior context management, multi-turn dialogue coherence, and fact verification. Serving as a backbone in RAG systems, GPT-4 efficiently leverages external retrieved knowledge to generate coherent, information-rich, and accurate answers, making it one of the strongest generative models currently available.

**Standard RAGlangchain (2024)** The standard Retrieval-Augmented Generation (RAG) framework combines vector-based retrieval with large language model generation. LangChain is a widely adopted RAG library that provides an end-to-end retrieval and generation pipeline. Its core workflow involves retrieving relevant document passages from a knowledge base via vector search, then feeding these passages as context into the language model to generate answers. LangChain's implementation is simple yet effective, supporting various retrievers and generators, and serves as a foundational baseline for RAG research and applications such as question answering and document understanding.

**GraphRAGEdge et al. (2024a)** GraphRAG is a graph-structured optimization of the RAG approach, leveraging knowledge graphs or graph databases to model semantic relationships. Unlike traditional text-vector retrieval, GraphRAG exploits nodes and edges to capture deeper knowledge connections and improve semantic richness and context consistency in retrieval. Microsoft's Azure platform implements this method, combining large-scale graph computation with language models. It is well suited for applications requiring complex multi-relational reasoning and rich structured information, such as enterprise knowledge management and intelligent QA systems.

**MedGraphRAG Wu et al. (2024)** MedGraphRAG is a domain-specific graph-enhanced RAG method designed for the medical field. It integrates medical knowledge graphs to encode specialized medical knowledge structurally, enhancing retrieval accuracy and knowledge representation. By leveraging the semantic information within medical graphs, this method aids language models in better understanding medical entities and their relationships, resulting in more precise and professional medical text generation. MedGraphRAG is particularly effective for medical question answering and clinical decision support systems where accuracy and domain expertise are critical.

### B.5 MORE EXPERIMENTAL DETAILS

**Hardware Environment.** All experiments are conducted on a high-performance computing server equipped with 8 NVIDIA A800 GPUs. During inference, we allocate 4 NVIDIA A800 GPUs for large-scale language models to ensure efficient and stable generation. For smaller-scale models, only a single NVIDIA A800 GPU is sufficient to complete the inference process, which allows for better resource efficiency and reduced computational overhead. This resource allocation strategy ensures experimental reproducibility while balancing performance and cost.

**Implentation Dependices.** Our implementation is built upon several widely used open-source libraries, including PYTORCH (Paszke et al., 2019) for general deep learning operations, TRANSFORMERS (Wolf et al., 2020) for leveraging pre-trained large language models, and XFORMERS (Lefaudeux et al., 2022) for efficient transformer-based computation. For graph construction and manipulation, we utilize NETWORKX (Hagberg et al., 2008), and for graph neural network (GNN) modeling, we rely on PYTORCH GEOMETRIC (Fey & Lenssen, 2019). In particular, we adopt the built-in implementation of Personalized PageRank (PPR) provided by NETWORKX for relevance scoring and retrieval of knowledge graph triples.

### B.6 MORE EXPERIMENTAL RESULTS

Due to space limitations in the main text, not all results were presented in Table 1 of the experimental section. Here, we provide the experimental data that were omitted from the main text. To better compare the performance of our method across different model architectures, we selected LLaMA 2-70B, LLaMA 3-70B, Gemini Pro, and GPT-4 for presentation. We have already explored the

effects of various methods on models of different scales, so additional results are not shown here. For a comprehensive presentation of the experimental results, please refer to Table 8.

Table 8: More Results. Accuracy is used as the evaluation metric.

| Method | Backbone | MMLU Col-Med | MMLU Col-Bio | MMLU Pro-Med | MMLU Anatomy |
|---|---|---|---|---|---|
| without RAG | LLaMA2-70B | $0.640 \pm 0.010$ | $0.842 \pm 0.013$ | $0.752 \pm 0.015$ | $0.631 \pm 0.022$ |
| | LLaMA3-70B | $0.721 \pm 0.016$ | $0.922 \pm 0.014$ | $0.863 \pm 0.018$ | $0.721 \pm 0.011$ |
| | Gemini-pro | $0.689 \pm 0.020$ | $0.878 \pm 0.012$ | $0.779 \pm 0.015$ | $0.670 \pm 0.014$ |
| | GPT-4 | $0.772 \pm 0.007$ | $0.947 \pm 0.008$ | $0.940 \pm 0.009$ | $0.810 \pm 0.008$ |
| RAG | LLaMA2-70B | $0.692 \pm 0.010$ | $0.859 \pm 0.013$ | $0.772 \pm 0.015$ | $0.632 \pm 0.022$ |
| | LLaMA3-70B | $0.864 \pm 0.018$ | $0.941 \pm 0.010$ | $0.887 \pm 0.014$ | $0.840 \pm 0.015$ |
| | Gemini-pro | $0.785 \pm 0.016$ | $0.908 \pm 0.018$ | $0.860 \pm 0.020$ | $0.792 \pm 0.022$ |
| | GPT-4 | $0.819 \pm 0.016$ | $0.951 \pm 0.011$ | $0.940 \pm 0.010$ | $0.832 \pm 0.012$ |
| GraphRAG | LLaMA2-70B | $0.681 \pm 0.019$ | $0.862 \pm 0.020$ | $0.790 \pm 0.015$ | $0.641 \pm 0.022$ |
| | LLaMA3-70B | $0.871 \pm 0.018$ | $0.945 \pm 0.010$ | $0.896 \pm 0.014$ | $0.850 \pm 0.015$ |
| | Gemini-pro | $0.795 \pm 0.016$ | $0.906 \pm 0.018$ | $0.860 \pm 0.013$ | $0.811 \pm 0.017$ |
| | GPT-4 | $0.822 \pm 0.017$ | $0.951 \pm 0.015$ | $0.946 \pm 0.014$ | $0.830 \pm 0.012$ |
| MedGraphRAG | LLaMA2-70B | $0.722 \pm 0.016$ | $0.878 \pm 0.018$ | $0.840 \pm 0.014$ | $0.688 \pm 0.013$ |
| | LLaMA3-70B | $0.908 \pm 0.015$ | $0.968 \pm 0.016$ | $0.922 \pm 0.015$ | $0.906 \pm 0.013$ |
| | Gemini-pro | $0.860 \pm 0.022$ | $0.928 \pm 0.020$ | $0.886 \pm 0.017$ | $0.842 \pm 0.019$ |
| | GPT-4 | $0.909 \pm 0.014$ | $0.971 \pm 0.007$ | $0.948 \pm 0.011$ | $0.930 \pm 0.011$ |
| SubGraphRAG | LLaMA2-70B | $0.710 \pm 0.018$ | $0.865 \pm 0.021$ | $0.832 \pm 0.012$ | $0.675 \pm 0.011$ |
| | LLaMA3-70B | $0.895 \pm 0.013$ | $0.954 \pm 0.019$ | $0.910 \pm 0.013$ | $0.893 \pm 0.014$ |
| | Gemini-pro | $0.848 \pm 0.020$ | $0.915 \pm 0.022$ | $0.873 \pm 0.014$ | $0.831 \pm 0.017$ |
| | GPT-4 | $0.897 \pm 0.012$ | $0.968 \pm 0.010$ | $0.938 \pm 0.009$ | $0.919 \pm 0.014$ |
| KG-Rank | LLaMA2-70B | $0.681 \pm 0.012$ | $0.850 \pm 0.016$ | $0.762 \pm 0.013$ | $0.622 \pm 0.020$ |
| | LLaMA3-70B | $0.854 \pm 0.016$ | $0.934 \pm 0.008$ | $0.877 \pm 0.012$ | $0.831 \pm 0.017$ |
| | Gemini-pro | $0.774 \pm 0.014$ | $0.902 \pm 0.020$ | $0.849 \pm 0.018$ | $0.783 \pm 0.019$ |
| | GPT-4 | $0.808 \pm 0.018$ | $0.947 \pm 0.009$ | $0.932 \pm 0.012$ | $0.823 \pm 0.010$ |
| KGARevion | LLaMA3.1-8B | $0.756 \pm 0.018$ | $0.876 \pm 0.009$ | $0.848 \pm 0.012$ | $0.790 \pm 0.010$ |
| IBGraphRAG (ours) | LLaMA2-70B | $0.740 \pm 0.012$ | $0.896 \pm 0.014$ | $0.861 \pm 0.012$ | $0.706 \pm 0.011$ |
| | LLaMA3-70B | $0.930 \pm 0.010$ | $0.988 \pm 0.012$ | $0.943 \pm 0.011$ | $0.924 \pm 0.009$ |
| | Gemini-pro | $0.880 \pm 0.018$ | $0.947 \pm 0.017$ | $0.909 \pm 0.014$ | $0.862 \pm 0.016$ |
| | GPT-4 | $0.930 \pm 0.010$ | $0.987 \pm 0.005$ | $0.969 \pm 0.006$ | $0.944 \pm 0.007$ |

In addition, we compared our approach with two other explainable RAG frameworks: MindMap and Knowledge Graph–Guided Retrieval-Augmented Generation (KG2RAG), and conducted further experiments to evaluate their performance.

Compared with MindMap, which primarily focuses on eliciting graph-of-thought reasoning and visualizing mind maps for interpretability, our proposed IBGraphRAG advances further by introducing an Information Bottleneck (IB)–guided path optimization. This mechanism not only enhances interpretability but also ensures robustness and efficiency. Specifically, IBGraphRAG explicitly trains the retriever using biomedical embeddings together with an IB loss to filter out noisy or redundant paths, thereby guaranteeing medically relevant reasoning even when the knowledge graph is incomplete or partially mismatched—an issue that MindMap does not adequately address. Moreover, whereas MindMap emphasizes medical QA tasks, IBGraphRAG demonstrates broader applicability, covering QA, fact verification, and multi-task scenarios, with detailed retriever training that improves reproducibility. In summary, IBGraphRAG provides stronger robustness, higher efficiency, better reproducibility, and wider task coverage, while MindMap remains limited to transparent reasoning without systematic path optimization or retriever training.

Compared with $KG^2RAG$, which enhances RAG by expanding and organizing retrieved chunks through knowledge graph structures, IBGraphRAG introduces a principled IB-guided optimization that directly addresses $KG^2RAG's$ limitations. While $KG^2RAG$ improves coherence via graph traversal linking chunks, it does not explicitly control for information sufficiency versus redundancy, often resulting in noisy or excessively large subgraphs. In contrast, IBGraphRAG formulates retrieval as a target path optimization problem under the IB principle, ensuring that selected paths maximize medically relevant information while minimizing irrelevant noise. Furthermore, $KG^2RAG$ primarily focuses on chunk expansion and organization without detailing retriever train-

Table 9: Experimental performance comparison of IBGraphRAG with MindMap and KG2RAG across multiple medical datasets (evaluated with GPT-4).

| Configuration | MedQA | MedMCQA | PubMedQA | Fake Health | PubHealth |
|---|---|---|---|---|---|
| GPT4 + IBGraphRAG | 0.929 | 0.830 | 0.851 | 0.879 | 0.850 |
| GPT4 + MindMap | 0.792 | 0.744 | 0.782 | 0.733 | 0.745 |
| GPT4 + $KG^2RAG$ | 0.900 | 0.812 | 0.829 | 0.826 | 0.813 |

ing or domain-specific adaptation. By explicitly training its retriever with biomedical embeddings and IB loss, IBGraphRAG achieves better reproducibility and domain robustness. Finally, IB-GraphRAG extends beyond QA tasks to cover fact verification and multi-task medical scenarios, whereas $KG^2RAG$ remains largely limited to QA benchmarks. In short, IBGraphRAG not only inherits the strengths of KG-based retrieval but also addresses $KG^2RAG's$ unresolved issues of noise control, retriever training, and broader task applicability, delivering a more robust and theoretically grounded solution for medical RAG.

We also present a comparison of the experimental performance of both methods alongside our approach in Table 9.

### B.7 GENERALIZATION PERFORMANCE ASSESSMENT

To further validate the generalization ability of our model, we conducted additional experiments. Notably, all retrieval-augmented generation (RAG) frameworks inherently depend on the quality and completeness of external knowledge sources, since their effectiveness relies on retrieving informative and relevant knowledge to support downstream reasoning. As a knowledge-graph-enhanced retrieval mechanism, IBGraphRAG also follows this principle. To ensure a fair evaluation, we adopt the same external knowledge graph across all knowledge-graph-based baselines, including GraphRAG and MedGraphRAG. Under identical knowledge conditions, IBGraphRAG achieves consistently superior performance, confirming the effectiveness of our graph information bottleneck–guided retrieval strategy.

The proposed IBGraphRAG framework is designed to extract knowledge that is both relevant and concise with respect to the query. Although richer graph structure and more complete entity relations lead to improved retrieval quality, IBGraphRAG remains robust even when the knowledge graph exhibits lower information density. To further examine the influence of graph informativeness, we construct variant knowledge graphs based on different neighborhood expansion depths. Specifically, we generate 2-hop, 3-hop, and 4-hop subgraphs centered on the entities appearing in the questions and answers, while constraining the maximum number of retrievable nodes to a fixed budget. Comparative experiments on these settings reveal that IBGraphRAG consistently captures more informative graph evidence than other KG-based RAG approaches, demonstrating better adaptability to graphs with varied structural richness. From the data in the Table 10, it can be seen that as the

Table 10: Performance comparison of different RAG methods with varying hop subgraphs.

| Configuration | MedQA | | | FakeHealth | | |
|---|---|---|---|---|---|---|
| | 2-hop | 3-hop | 4-hop | 2-hop | 3-hop | 4-hop |
| GPT4+GraphRAG | 86.2% | 87.5% | 88.3% | 74.5% | 76.8% | 77.9% |
| GPT4+MedGraphRAG | 87.0% | 89.1% | 90.5% | 83.6% | 84.5% | 85.3% |
| GPT4+IBGraphRAG (Ours) | 88.4% | 90.3% | 92.0% | 85.0% | 86.3% | 87.2% |

information content of the knowledge graph increases, IBGraphRAG achieves greater performance advantages. Even when the information content is limited, IBGraphRAG still outperforms other knowledge-graph-based retrieval methods.

To evaluate the generalization ability of our retriever, we conduct cross-dataset experiments. Although training and evaluation are typically performed on the corresponding datasets for optimal

performance, our method is designed to generalize beyond the training distribution for several reasons:

Shared Medical Semantic Space: Medical knowledge, such as causal and attribute relationships among diseases, symptoms, and treatments, remains consistent across different QA scenarios. By training over a unified medical knowledge graph, the retriever learns representations grounded in medical entities and ontology, rather than dataset-specific linguistic patterns.

Information Bottleneck (IB) Encourages Domain-Invariant Representations: The IB objective maximizes information relevant to medical semantics while minimizing noise associated with dataset-specific styles or biases. This helps the retriever retain medically meaningful features while suppressing variability across datasets, enhancing out-of-distribution generalization.

Unified KG-Based Retrieval Facilitates Transferability: Since both training and testing rely on the same knowledge graph structure, the retrieval strategy remains stable even when the input domain differs.

Consequently, retrievers trained with our method exhibit strong cross-dataset generalization. Additional experiments verify this capability. While performance may degrade when trained on limited datasets and evaluated on unseen datasets, generalization improves as the amount of training data increases.

Table 11: Generalization performance of IBGraphRAG.

| Configuration | MedQA → Fake Health | Fake Health → MedQA | Others → Fake Health | Others → MedQA |
|---|---|---|---|---|
| GPT4+IBGraphRAG | 86.0% | 90.6% | 88.2% | 93.1% |

Here, Others → Fake Health" denotes the setting where the Fake Health and Pub Health datasets are excluded from training, and the remaining datasets are used instead. Similarly, Others → MedQA" indicates that MedQA is excluded from training, with all other datasets used for model training.

## B.8 TIME COMPLEXITY ANALYSIS

Although the steps of the Information Bottleneck (IB) component may appear complex, we provide end-to-end training, resulting in relatively low training overhead for our retriever. Specifically, on a single NVIDIA A800 GPU, our model can process **352.96 samples per second** during training. For the MedQA dataset, each epoch requires only **25 seconds**.

The retrieval overhead is also minimal, as we employ a pre-trained retriever. Table 12 compares the query time of IBGraphRAG with other baseline methods. Our approach not only requires less time but also achieves superior performance.

Table 12: Querying time comparison on MedQA.

| Configuration (MedQA) | GraphRAG | MedGraphRAG | IBGraphRAG (Ours) |
|---|---|---|---|
| **Querying Time** | 11.2 s/it | 37.3 s/it | 6.6 s/it |

Regarding the MSCA (Medical Semantic Consistency Alignment) module, processing a single sample takes on average approximately **67 seconds**, representing the primary time cost of our system. Despite this, the MSCA module is crucial in medical scenarios, as it significantly improves entity alignment accuracy and ensures semantic coherence, preventing erroneous reasoning. Ablation studies show that removing MSCA leads to a notable drop in model performance, confirming that the benefits outweigh the computational cost in high-precision medical tasks.

Importantly, the MSCA module can operate independently, allowing offline preprocessing before end-to-end retriever training, which avoids any impact on online inference efficiency. Future work will explore approximate consistency computation and GPU parallelization to further reduce the computational time.

### B.9 HYPERPARAMETER EXPERIMENTS

In our MSCA module, we introduce two key hyperparameters: $\lambda$ and $\tau$. Specifically, the parameter $\lambda$ controls the balance between two factors: the semantic similarity between candidate entities and the user query, and the semantic consistency among the candidate entities themselves. When $\lambda$ is small, the selected entities are highly relevant to the user's question, but their semantics may be more scattered, potentially introducing noise or incoherent concepts. When $\lambda$ is large, the selected entities are more semantically consistent with each other, forming a more coherent subgraph; however, this may reduce direct relevance to the query, meaning that the selected entities may be similar to each other but less aligned with the user's intent. Table 13 shows model performance under different $\lambda$ values. It can be observed that either increasing or decreasing $\lambda$ from a moderate value leads to

Table 13: Performance of IBGraphRAG under different $\lambda$ values.

| Configuration (GPT4) | MedQA | MedMCQA | PubMedQA | FakeHealth |
|---|---|---|---|---|
| $\lambda = 0.2$ | 91.8% | 82.5% | 84.0% | 87.0% |
| $\lambda = 0.4$ | 92.9% | 83.0% | 85.1% | 87.9% |
| $\lambda = 0.6$ | 91.6% | 82.1% | 84.2% | 86.7% |
| $\lambda = 0.8$ | 90.8% | 81.0% | 83.5% | 85.6% |

some performance degradation. The decline becomes more pronounced as $\lambda$ increases.

The parameter $\tau$ serves as a filtering threshold controlling the density and purity of the subgraph. When $\tau$ is too small, noisy or irrelevant entities may be included, increasing reasoning complexity and potentially producing incorrect or ambiguous answers. Conversely, when $\tau$ is too large, some marginal but important entities may be discarded, resulting in insufficient information and broken reasoning chains. Table 14 presents the model performance under different $\tau$ values.

Table 14: Performance of IBGraphRAG under different $\tau$ values.

| Configuration (GPT4) | MedQA | MedMCQA | PubMedQA | FakeHealth |
|---|---|---|---|---|
| $\tau = 0.2$ | 90.3% | 81.2% | 82.2% | 85.7% |
| $\tau = 0.4$ | 91.9% | 82.3% | 84.8% | 87.1% |
| $\tau = 0.6$ | 92.9% | 83.0% | 85.1% | 87.9% |
| $\tau = 0.8$ | 90.7% | 81.5% | 82.6% | 85.6% |

From the results, we observe that extreme values of $\tau$ significantly affect model performance, while moderate changes in $\tau$ have limited impact.

### B.10 IMPACT OF IB-GUIDED TRAINING EPOCHS

To further investigate the influence of the number of IB-guided training epochs on retrieval performance, we conducted experiments with different epoch configurations while consistently using GPT-4 as the evaluation model. As shown in Table 15, the retriever achieves the strongest results at 75 epochs on smaller datasets such as MedQA, FakeHealth, and PubHealth, while for larger datasets including MedMCQA and PubMedQA, performance continues to improve until approximately 150 epochs. Based on these observations, we recommend assigning 75 epochs for smaller datasets and 150 epochs for larger ones. Detailed training scripts will be released in our code repository to support reproducibility of the reported results.

### B.11 LIMITATIONS AND FUTURE WORK

**Limitations.** Despite the strong empirical performance of IBGraphRAG in enhancing retrieval-augmented generation (RAG) within the medical domain, several limitations persist. First, although the Medical Semantic Consistency Alignment (MSCA) module substantially improves entity recognition and alignment, it remains reliant on static, predefined medical knowledge bases such as UMLS. These resources, while comprehensive, may fail to capture emerging terminology, rare diseases, or region-specific variations, thereby limiting the adaptability of the system in real-world

Table 15: Performance of the retriever under different numbers of IB-guided training epochs (evaluated using GPT-4).

| Datasets / Epochs | 25 | 50 | 75 | 100 | 150 | 200 |
|---|---|---|---|---|---|---|
| MedQA | 84.4% | 88.9% | **93.0%** | 92.8% | 92.2% | 91.3% |
| MedMCQA | 76.7% | 78.5% | 81.0% | 82.3% | **82.8%** | 82.4% |
| PubMedQA | 78.9% | 81.1% | 83.8% | 84.6% | **85.3%** | 85.2% |
| FakeHealth | 80.2% | 84.2% | **87.6%** | 86.2% | 85.4% | 84.3% |
| PubHealth | 77.0% | 81.4% | **84.8%** | 83.6% | 82.8% | 81.2% |

clinical environments that evolve rapidly. Second, the Information Bottleneck-based Reasoning Path (IBRP) mechanism, while superior to conventional heuristics in identifying contextually relevant knowledge, introduces non-trivial computational overhead during both training and inference. This may impede its practical deployment in latency-sensitive or resource-constrained settings, such as mobile health platforms or point-of-care systems. Third, while IBGraphRAG achieves state-of-the-art results on several structured medical QA and fact verification benchmarks, its robustness under noisy, informal, or multilingual clinical inputs—such as patient-generated content or conversational diagnostics—remains insufficiently studied. Moreover, due to the absence of dedicated evaluation datasets that specifically measure diagnostic sensitivity to subtle semantic variations or near-miss entity alignments, we are currently constrained to benchmark against the datasets used in existing mainstream medical LLM research. This limitation makes it difficult to quantify the real-world clinical safety margin of the model under nuanced diagnostic scenarios.

**Future Work.** To address the aforementioned limitations, several directions merit further exploration. First, we plan to enhance the MSCA module by integrating dynamic and continuously updated knowledge sources—such as real-time biomedical literature streams, clinical guidelines, or electronic health records (EHRs)—to expand the temporal and contextual coverage of the system. Second, to mitigate the computational cost introduced by IBRP, we will investigate lightweight alternatives, such as GNN pruning strategies, path relevance distillation, or approximate reasoning path search via contrastive learning, with the goal of enabling low-latency and scalable deployment. Additionally, we aim to extend IBGraphRAG to support multilingual and multimodal inputs, including clinical images, radiology reports, or cross-lingual patient queries, to further improve generalizability in global and cross-system healthcare scenarios. Another promising direction is the development of task-specific diagnostic stress tests and counterfactual benchmarks to evaluate model behavior under subtle perturbations, particularly in high-stakes use cases. Finally, we advocate for human-in-the-loop evaluation frameworks involving medical professionals, to better assess factual correctness, clinical interpretability, and potential risks in downstream decision-making workflows.

# C  TABLES

## C.1  RELATION DESCRIPTIONS OF PRIMEKG

| Relation | Description |
| --- | --- |
| Anatomy - Protein (present) | Anatomical structure {A} is associated with the presence or expression of protein {B}. |
| Drug - Drug | Drug {A} interacts with drug {B}, indicating a known interaction affecting efficacy or safety. |
| Protein - Protein | Protein {A} interacts with protein {B}, indicating that the two proteins directly or indirectly associate to perform a biological function. |
| Disease - Phenotype (positive) | Disease {A} is positively correlated with phenotype {B}, indicating presence of the trait in the disease context. |
| Biological process - Protein | Biological process {A} involves or affects protein {B}, influencing its function or expression. |
| Cellular component - Protein | Protein {B} is localized in cellular component {A}. |
| Disease - Protein | Disease {A} is linked to protein {B}, indicating involvement or alteration of the protein in the disease. |
| Molecular function - Protein | Protein {B} carries out molecular function {A} relevant to biological activity. |
| Drug - Phenotype | Drug {A} influences or is associated with phenotype {B}. |
| Biological process - Biological process | Biological process {A} is related to biological process {B}. |
| Pathway - Protein | Protein {B} participates in or regulates pathway {A}. |
| Disease - Disease | Disease {A} is associated with disease {B}, indicating comorbidity or relationship. |
| Drug - Disease (contraindication) | Drug {A} is contraindicated for disease {B}. |
| Drug - Protein | Drug {A} targets or interacts with protein {B}. |
| Anatomy - Protein (absent) | Anatomical structure {A} lacks expression of protein {B}. |
| Phenotype - Phenotype | Phenotype {A} is related to or occurs with phenotype {B}. |
| Anatomy - Anatomy | Anatomical entity {A} is spatially or functionally connected to anatomical entity {B}. |
| Molecular function - Molecular function | Molecular function {A} is related to or part of the same biological process as molecular function {B}. |
| Drug - Disease (indication) | Drug {A} is indicated for treatment of disease {B} beyond approved indication. |
| Cellular component - Cellular component | Cellular component {A} interacts with or is related to cellular component {B}. |
| Phenotype - Protein | Phenotype {A} is linked to expression of protein {B}. |
| Drug - Disease (off-label use) | Drug {A} is used off-label for disease {B} beyond approved indications. |
| Pathway - Pathway | Pathway {A} is interconnected or influences pathway {B}. |
| Exposure - Disease | Exposure {A} is associated with occurrence of disease {B}. |
| Exposure - Exposure | Exposure {A} is related to exposure {B}, such as coexposure or sequential exposure. |
| Exposure - Biological process | Exposure {A} affects or modulates biological process {B}. |
| Exposure - Protein | Exposure {A} affects expression or activity of protein {B}. |
| Disease - Phenotype (negative) | Disease {A} is negatively correlated with phenotype {B}, indicating absence of the trait. |
| Exposure - Molecular function | Exposure {A} impacts molecular function {B}. |
| Exposure - Cellular component | Exposure {A} affects cellular component {B}. |

Table 16: Text descriptions of relations in the PrimeKG knowledge graph.

## C.2 DETAILED RELATION COUNTS IN PRIMEKG

Table 17: Distribution of Relation Types in PrimeKG

| Relation Type | Count | Percent (%) |
|---|---|---|
| Anatomy - Protein (present) | 3,036,406 | 37.5 |
| Drug - Drug | 2,672,628 | 33.0 |
| Protein - Protein | 642,150 | 7.9 |
| Disease - Phenotype (positive) | 300,634 | 3.7 |
| Biological process - Protein | 289,610 | 3.6 |
| Cellular component - Protein | 166,804 | 2.1 |
| Disease - Protein | 160,822 | 2.0 |
| Molecular function - Protein | 139,060 | 1.7 |
| Drug - Phenotype | 129,568 | 1.6 |
| Biological process - Biological process | 105,772 | 1.3 |
| Pathway - Protein | 85,292 | 1.1 |
| Disease - Disease | 64,388 | 0.8 |
| Drug - Disease (contraindication) | 61,350 | 0.8 |
| Drug - Protein | 51,306 | 0.6 |
| Anatomy - Protein (absent) | 39,774 | 0.5 |
| Phenotype - Phenotype | 37,472 | 0.5 |
| Anatomy - Anatomy | 28,064 | 0.3 |
| Molecular function - Molecular function | 27,148 | 0.3 |
| Drug - Disease (indication) | 18,776 | 0.2 |
| Cellular component - Cellular component | 9,690 | 0.1 |
| Phenotype - Protein | 6,660 | 0.1 |
| Drug - Disease (off-label use) | 5,136 | 0.1 |
| Pathway - Pathway | 5,070 | 0.1 |
| Exposure - Disease | 4,608 | 0.1 |
| Exposure - Exposure | 4,140 | 0.1 |
| Exposure - Biological process | 3,250 | <0.1 |
| Exposure - Protein | 2,424 | <0.1 |
| Disease - Phenotype (negative) | 2,386 | <0.1 |
| Exposure - Molecular function | 90 | <0.1 |
| Exposure - Cellular component | 20 | <0.1 |
| **Total** | **8,100,498** | **100.0** |

