# OpenReview forum: "IBGraphRAG: Enhancing Medical Knowledge Graph Retrieval Based on Semantic Consistency and Information Bottleneck"
_ICLR.cc/2026/Conference — ICLR 2026 Conference Withdrawn Submission_

### Official Review · Reviewer_ZVYq · 2025-10-28

**Soundness:** 3
**Presentation:** 3
**Contribution:** 3
**Rating:** 6
**Confidence:** 3

**Summary:**

The main contribution of this paper is a three-phase knowledge-graph-powered approach for medical question answering / fact verification: 1) use an LLM to extract entities from a user query, then augment with a medical knowledge base (this includes enriching with synonyms and encoding with BioWordVec); 2) information bottleneck over a query graph to find "reasoning paths" between a question and answer; 3) an LLM takes as input information retrieved from the knowledge graph to generate a final answer. Experiments over several QA and fact verification datasets show strong performance versus many baselines (and across many base LLMs).

**Strengths:**

+ The idea of bringing information bottleneck to this medical KG scenario makes a lot of sense.

+ The experimental results appear quite strong and comprehensive. There are also some good ablations to showcase the importance of the entity alignment and the IB component.

**Weaknesses:**

- The authors acknowledge in the limitations that the information bottleneck method results in "non-trivial computational overhead during both training and inference". It would be good to quantify this overhead. In particular, the method itself is quite complex (even beyond the IB part) with lots of steps.

- I believe the IB component is trained over known QA-pairs. Do you have any evidence on how the method performs "in-the-wild" in scenarios where there is not clean training data? Does it generalize from one dataset to another?

- There is some missing highly-relevant work -- KG-Rank: Enhancing Large Language Models for Medical QA with Knowledge Graphs and Ranking Techniques https://arxiv.org/abs/2403.05881 -- that also uses LLMS + KGs. The technique there uses ranking and no learning, but it would be worthwhile to discuss the connections.

- Figure 3 shows two graphs -- one with IB guidance and one without. It is not clear to me what we are meant to learn from this figure. How to interpret which nodes are highlights since we don't know the context? The discussion may be sufficient without the graph.

**Questions:**

See above

---

> ### Author Response · Authors · 2025-11-18
>
> Thank you for your feedback. We are greatly encouraged by your recognition of the soundness and novelty of our ideas, as well as the performance of our experiments. Regarding the shortcomings and limitations you raised, we will provide further explanations.
>
>
> > **W1.** *The authors acknowledge in the limitations that the information bottleneck method results in "non-trivial computational overhead during both training and inference". It would be good to quantify this overhead. In particular, the method itself is quite complex (even beyond the IB part) with lots of steps.
>
> Thank you for your feedback. We have added specific experimental data to help quantify the computational overhead of our method.
>
> In fact, although the steps of the IB component may appear complex in the paper, we provide end-to-end training, so the actual training overhead of our retriever is not large. Specifically, on a single A800 GPU, it can process **352.96 samples per second** during training, and for MedQA, each epoch takes only **25 seconds**. Regarding the specific retrieval overhead, we have added comparisons with other baselines. As our experiments are conducted using a trained retriever, the retrieval overhead is extremely low. Compared to other baselines, our method not only requires less time but also achieves better performance.
>
> | Configuration(Medqa) | GraphRAG | MedGraphRAG | IBGraphRAG(Ours) |
> | -------------------- | -------- | ----------- | ---------------- |
> | **Querying Time** | 11.2s/it | 37.3s/it | 6.6s/it|
>
> Regarding the MSCA (Medical Semantic Consistency Alignment) module in our paper, our tests indicate that processing a single sample takes approximately **67 seconds** on average, representing the main time cost of our system. However, this design is particularly important in medical scenarios, as it substantially improves the accuracy of entity alignment and ensures semantic coherence, thereby helping to prevent erroneous reasoning. Ablation experiments further show that removing MSCA leads to a significant drop in model accuracy, demonstrating that its benefits outweigh the computational cost, especially in the medical domain, which requires high precision..
>
> **Notably, the MSCA module can operate independently, allowing us to preprocess data offline before end-to-end retriever training, thereby avoiding any impact on online inference efficiency. In future work, we also plan to explore approximate consistency computation and GPU parallelization to further reduce the time cost.**
>
>
> > **W2.** *I believe the IB component is trained over known QA-pairs. Do you have any evidence on how the method performs "in-the-wild" in scenarios where there is not clean training data? Does it generalize from one dataset to another?
>
> Thank you for your insightful question. To achieve optimal training performance, we typically train and evaluate the retriever on the corresponding datasets. However, we emphasize that our method is designed to generalize beyond the training distribution due to the following reasons:
>
> 1. **Shared medical semantic space across datasets**
>     Medical knowledge (e.g., causal and attribute relationships among diseases, symptoms, and treatments) remains consistent across different QA scenarios. Our retriever is trained over a unified medical knowledge graph, thereby learning semantics grounded in medical entities and ontology rather than dataset-specific linguistic patterns.
> 2. **Information Bottleneck (IB) promotes domain-invariant representations**
>     The IB objective encourages the model to:
>     - maximize information relevant to medical semantics
>     - minimize noise tied to dataset-specific writing styles or biases
>         This helps the retriever retain medical commonality while suppressing dataset variability, enabling stronger out-of-distribution generalization.
> 3. **Unified KG-based retrieval enables natural transferability**
>     Since both training and testing rely on the same knowledge graph structure, the retrieval strategy remains stable even when the input domain changes.
>
> As a result, retrievers trained with our method demonstrate strong cross-dataset generalization ability. We have also added additional experiments to verify this capability. While performance may drop when trained on limited data and evaluated on unseen datasets, we observe that the generalization ability improves as the amount of training data increases.
>
> | Configuration       | Medqa->Fake Health | Fake Health->Medqa | Others->Fake Health | Others->Medqa |
> | ------------------- | ------------------ | ------------------ | ------------------- | ------------- |
> | **GPT4+IBGraphRAG** | 0.860 | 0.906 | 0.882| 0.931|
>
> Here, 'Others → Fake Health' refers to the setting where the Fake Health and Pub Health datasets are excluded from training, and other datasets are used instead. Similarly, 'Others → MedQA' refers to the setting where the MedQA dataset is excluded and other datasets are used for training.

---

> > ### Author Response · Authors · 2025-11-18
> >
> > > **W3.** *There is some missing highly-relevant work -- KG-Rank: Enhancing Large Language Models for Medical QA with Knowledge Graphs and Ranking Techniques [https://arxiv.org/abs/2403.05881](https://arxiv.org/abs/2403.05881) -- that also uses LLMS + KGs. The technique there uses ranking and no learning, but it would be worthwhile to discuss the connections.
> >
> > Thank you for your feedback. To better highlight our contributions, we conducted a deeper comparison with KG-Rank.
> >
> > 1. Compared with KG-Rank, our proposed MSCA module not only considers query relevance but also takes into account the internal consistency of the candidate entity set, ensuring that the medical entities within the subgraph are more coherent and clinically logical. In contrast, KG-Rank mainly focuses on filtering out noise but does not guarantee semantic consistency.
> >
> > 2. Our proposed IBRP module is a trainable module that selects reasoning paths based on the information bottleneck principle, retaining diagnosis-relevant information while compressing irrelevant information. This not only improves factual consistency but also generates reasoning chains that better align with medical causal logic, enhancing interpretability.
> >
> > 3. IBGraphRAG, by combining MSCA and IBRP, can intuitively demonstrate “why the model arrived at this answer,” which is particularly important in medical scenarios. In contrast, KG-Rank only provides ranking results, making it difficult to display a complete reasoning chain.
> > We also evaluated the performance of IBGraphRAG and KG-Rank on five datasets.
> >
> > | Configuration       | Medqa | MedMCQA | PubMedQA | Fake Health | PubHealth |
> > | ------------------- | ----- | ------- | -------- | ----------- | --------- |
> > | **GPT4+IBGraphRAG** | 0.929 | 0.830   | 0.851    | 0.879       | 0.850     |
> > | **GPT4+KG-Rank**    | 0.864 | 0.742   | 0.807    | 0.776       | 0.761     |
> >
> > > **W4.** *Figure 3 shows two graphs -- one with IB guidance and one without. It is not clear to me what we are meant to learn from this figure. How to interpret which nodes are highlights since we don't know the context? The discussion may be sufficient without the graph.
> >
> > We thank the reviewer for the comment on Figure 3. This figure is intended to illustrate the difference in retrieval path convergence with and without Information Bottleneck (IB) guidance, thereby validating that our method addresses the issue of insufficient diagnostic relevance in retrieved paths. Without IB guidance, the trained retriever tends to emphasize ambiguous or shallow connections (e.g., 'fever → chest discomfort'), whereas under IB guidance, it converges to medically coherent paths (e.g., 'cough → lower respiratory infection → alveolar congestion → pulmonary consolidation → pneumonia → chest pain'). The red highlighted nodes represent the auxiliary optimization target—the continuous shortest path between the question and answer entities, which the model is expected to learn in the absence of IB guidance—while the yellow nodes indicate the optimization process. This visualization demonstrates that, under IB guidance, the retriever is able to further optimize toward the auxiliary target, retrieving paths that are more clinically relevant and precise without needing to explicitly specify the exact path.

---

> > ### Comment · Reviewer_ZVYq · 2025-11-25
> >
> > Thank you for the clarifications.

---

> > > ### Author Response · Authors · 2025-11-26
> > >
> > > Thank you very much for your positive recognition of our work. We have carefully addressed your comments and provided detailed responses. We hope our revisions have adequately resolved your concerns. If there are any remaining questions or suggestions, we would be more than happy to respond.**If your concerns have been fully addressed, we sincerely look forward to your feedback and would greatly appreciate it if you could  improve your rating of our manuscript.**
> > >
> > > Thank you once again for your time and valuable input during the review process!

---

### Official Review · Reviewer_R3yy · 2025-10-31

**Soundness:** 2
**Presentation:** 3
**Contribution:** 2
**Rating:** 2
**Confidence:** 3

**Summary:**

IBGraphRAG introduces a medical-domain oriented RAG framework that integrates medical knowledge graphs to enhance factual accuracy and terminological precision in LLM-based QA. It contributes two main innovations: Medical Semantic Consistency Alignment, which improves entity recognition and linking by aligning LLM-extracted entities with the UMLS knowledge base using semantic consistency checks, and an Information Bottleneck-based Reasoning Path mechanism, which optimizes KG traversal to retain only contextually relevant, information-rich reasoning paths. Together, these components make IBGraphRAG better suited for medical reasoning, characterized by highly specialized reasoning and complex KG reasoning paths, achieving state-of-the-art results on multiple biomedical QA benchmarks.
While the results seem promising, the relationship with related work is not sufficiently clear, and it seems that core competitors are missing from the empirical evaluation.

**Strengths:**

- The proposed method demonstrates improved performance compared to current architectures.
	- The results are evaluated on 7 widely used question answering and fact verification tasks.
	- The proposed methodology seems sound and resonates with the need in scientific community of having highly efficient and scalable retriever component able to determine the paths in the graph that are relevant to a particular query.
	- The authors present ablation studies testing the importance of both of their introduced components

**Weaknesses:**

Major:
- W1 The paper could be better positioned with respect to existing work. Each paragraph in the related work section should clearly highlight how the proposed method addresses specific research gaps compared to prior approaches.
- W2 It is unclear what the exact relationship of the proposed method is with the existing SubGraphRAG (Li et al 2024a) - is it the same approach, applied to medical data? Why is it not part of the empirical evaluation?
- W3 Similarly, it is unclear how the proposed method compares with Su, Xiaorui, et al. "KGARevion: an AI agent for knowledge-intensive biomedical QA." ICLR 2025, or Yang, Rui, et al. "KG-Rank: Enhancing Large Language Models for Medical QA with Knowledge Graphs and Ranking Techniques." BioNLP 2024.

- The paper would benefit from a more detailed description - either in the main text or appendix - of the formulation of both tasks involved in the experimental setup, and how each task components are mapped to the IB-guided target path optimization process.
	- The analysis of temporal performance is missing. It would be useful to report time-related metrics, such as the duration of retriever training, the time required for generating training set alignments, and inference time.
	- Additional details about the UMLS KG would be needed—for example, the number of related terms and variants used to produce enriched entities within the MSCA component.
	- It is unclear how the retriever is trained or which retriever model is used.
	- An ablation study examining the impact of the number of IB-guidance epochs on performance would strengthen the paper. Currently, this is only shown for two graph examples in Figure 3.
	- Some statements are overly vague and should be clarified for reproducibility. For example:
		- (Line 178) "If relevant entities remain insufficient" - specify how this insufficiency is determined or quantified.
		- (Line 177) "A domain-specific medical model encodes" - clarify which model is used (e.g., BioWordVec?).
		- (Line 249) "Precompute semantic embeddings of all entities and relations in the knowledge graph using pre-trained text encoders" - specify which encoders are employed.
- (Line 109) "RAG... relatively underexplored in medical applications" seems a bit of a bold claim
	- Related work: The paper could be strengthened by discussing or benchmarking against:
		- Wen, Yilin, et al. "MindMap: Knowledge Graph Prompting Sparks Graph of Thoughts in Large Language Models."" ACL 2024.
		- Zhu, Xiangrong, et al. "Knowledge Graph-Guided Retrieval-Augmented Generation" NAACL 2025

Errata:
	- "semantics.and" --> line 165
	- "overlap,as" --> line 82
	- "Baselines with MSCA" to "Baselines without MSCA" --> line 441

**Questions:**

- W1-3 above

- (line 309) Recent studies on the information bottleneck --> which studies?
	- (line 173) "To train the retriever" - which retriever are you training? please provide additional details for reproducibility.
	- (line 178) "If relevant entities remain insufficient,", I think this is too vague, provide additional details on how this insufficiency is evaluated.
	- (line 177) "a domain-specific medical model encodes"
		- not clear which model exactly? BioWordVec?
	- (line 249) "precompute semantic embeddings of all entities and relations in the knowledge graph using pre-trained text encoders"
		- which ones?

---

> ### Author Response · Authors · 2025-11-18
>
> Thank you for recognizing the methodological soundness of our work, which aligns with the scientific community’s need for efficient and scalable retriever components, particularly in identifying the most relevant knowledge graph paths for a given query, as well as acknowledging the performance of our IBGraphRAG model. Your questions have been very helpful in improving the quality of our paper. We sincerely appreciate your suggestions and will address each of your comments and questions in detail.
>
> > **W1.** *The paper could be better positioned with respect to existing work. Each paragraph in the related work section should clearly highlight how the proposed method addresses specific research gaps compared to prior approaches.
>
> Thank you for your feedback. We have reorganized and supplemented the Related Work section so that each paragraph more clearly points out the limitations of existing methods and highlights the improvements that IBGraphRAG brings to the medical domain. The specific modifications are as follows:
>
> - **General RAG methods** [1],[3] perform well in open-domain tasks but lack fine-grained customization for clinical scenarios, resulting in insufficient terminological precision and factual consistency.
>
> - **Graph-based approaches** [2],[3] improve reasoning by leveraging knowledge graphs, but their retrieval strategies typically rely on heuristic algorithms, which are not always clinically informative.
>
> - **Medical-domain LLMs** [4-5] achieve strong reasoning ability through fine-tuning, but such approaches are costly and have limited accessibility, making them impractical for wide deployment.
>
>
> Building on these identified gaps, our work introduces two innovations:
>
> 1. **Medical Semantic Consistency Alignment (MSCA):** Addresses inaccurate entity recognition and alignment in medical queries by enforcing semantic consistency with structured medical knowledge bases.
>
> 2. **Information Bottleneck-based Reasoning Path (IBRP):** Prioritizes retrieval of clinically relevant reasoning paths rather than relying on heuristic algorithms, thereby enhancing diagnostic accuracy and interpretability.
>
> [1]Lewis P, Perez E, Piktus A, et al. Retrieval-augmented generation for knowledge-intensive nlp tasks[J]. Advances in neural information processing systems, 2020, 33: 9459-9474.
> [2]Edge D, Trinh H, Cheng N, et al. From local to global: A graph rag approach to query-focused summarization[J]. arXiv preprint arXiv:2404.16130, 2024.
> [3]Li M, Miao S, Li P. Simple is effective: The roles of graphs and large language models in knowledge-graph-based retrieval-augmented generation[J]. arXiv preprint arXiv:2410.20724, 2024.
> [4]Luo R, Sun L, Xia Y, et al. BioGPT: generative pre-trained transformer for biomedical text generation and mining[J]. Briefings in bioinformatics, 2022, 23(6): bbac409.
> [5]Singhal K, Tu T, Gottweis J, et al. Toward expert-level medical question answering with large language models[J]. Nature Medicine, 2025, 31(3): 943-950.
>
> > **W2.** *It is unclear what the exact relationship of the proposed method is with the existing SubGraphRAG (Li et al 2024a) - is it the same approach, applied to medical data? Why is it not part of the empirical evaluation?
>
> Thank you for your detailed question. In the revised manuscript, we have further clarified the relationship between IBGraphRAG and SubGraphRAG:
>
> SubGraphRAG proposes a general knowledge-graph retrieval-augmented framework, with its core being the efficient retrieval of subgraphs using a lightweight MLP combined with Directional Distance Encoding (DDE), followed by reasoning with an LLM. Its goal is to improve the efficiency and accuracy of general KGQA tasks.
>
> In contrast, IBGraphRAG focuses on medical knowledge graph scenarios and introduces two key innovations tailored to the medical domain:
>
> - **MSCA:** Addresses the lack of semantic consistency in medical terminology during KG retrieval, preventing the introduction of semantically biased entities.
>
> - **IBRP:** Retains the most diagnostically relevant reasoning paths in medical diagnostic inference, rather than relying solely on the shortest path or arbitrary subgraphs, thereby enhancing both interpretability and accuracy in medical question answering.
>
>
> Since SubGraphRAG’s experiments are mainly based on general KGQA benchmarks (e.g., WebQSP, CWQ), while our work focuses on medical knowledge graphs such as UMLS and DrugBank, the task definitions and data distributions differ significantly. A direct comparison would therefore be unfair, which is why we did not initially conduct a dedicated evaluation. Here, we have supplemented the manuscript with an evaluation of SubGraphRAG applied to medical datasets.
>
> | Configuration(GPT4) | Medqa | MedMCQA | PubMedQA | Fake Health | Pub Health |
> | ------------------- | ----- | ------- | -------- | ----------- | ---------- |
> | IBGraphRAG| 92.9% | 83.0%   | 85.1%    | 87.9%  | 85.0%  |
> | SubGraphRAG| 90.2% | 80.4% | 82.9% | 85.1% | 83.2% |

---

> > ### Author Response · Authors · 2025-11-18
> >
> > > **W3.** *Similarly, it is unclear how the proposed method compares with Su, Xiaorui, et al. "KGARevion: an AI agent for knowledge-intensive biomedical QA." ICLR 2025, or Yang, Rui, et al. "KG-Rank: Enhancing Large Language Models for Medical QA with Knowledge Graphs and Ranking Techniques." BioNLP 2024.
> >
> > Thank you for your feedback. We have added a detailed explanation to clarify the specific differences between KGARevion, KG-Rank, and our proposed IBGraphRAG.
> >
> > ### **Comparison with KGARevion**
> >
> > **1. Semantic Consistency**
> > - KGAREVION mainly relies on the structure of the knowledge graph to verify triple correctness.
> > - IBGraphRAG explicitly ensures strict semantic alignment between medical entities and structured knowledge through the MSCA module, effectively reducing semantic mismatches.
> >
> > **2. Reasoning Path Selection**
> > - KGAREVION emphasizes a generate–verify–revise workflow, while its path selection depends largely on structural validation.
> > - IBGraphRAG introduces the IBRP principle to highlight clinically relevant reasoning while compressing redundant information, leading to reasoning chains that better reflect real-world medical decision logic.
> >
> >  **3. Resource Consumption**
> > - KGARevion requires a fully fine-tuned LLM to support its entire workflow, which leads to **significantly higher computational resource usage**.
> > - IBGraphRAG, however, maintains strong performance **without any LLM fine-tuning**, making it more cost-effective and practical for broader medical deployment — which is one of the key motivations behind our design.
> > ### **Comparison with KG-Rank**
> >
> > **1. Semantic Consistency**
> > - KG-Rank enhances factual correctness by ranking and re-ranking retrieved triples, thereby improving the reliability of generated answers. However, it does not specifically address the semantic alignment of medical entities, which may result in potential semantic discrepancies.
> > - In contrast, IBGraphRAG introduces the **Medical Semantic Consistency Alignment (MSCA) module**, which strictly ensures consistency between entities and their semantics, reducing potential mismatches and improving the internal consistency of retrieved subgraphs.
> >
> > **2. Reasoning Path Selection**
> > - KG-Rank selects reasoning paths based on similarity and diversity, which is effective for filtering noise but may not fully capture clinical reasoning logic.
> > - IBGraphRAG applies the **Information Bottleneck Reasoning Path (IBRP) principle**, which compresses redundant information while emphasizing clinically relevant connections, thereby generating reasoning chains that better align with medical decision-making processes.
> >
> > **3. Interpretability**
> > - KG-Rank provides ranked triples for answer generation, enhancing factual accuracy, but offers limited transparency regarding the reasoning process.
> > - IBGraphRAG extends this by producing coherent reasoning chains in addition to answers, allowing clinicians to more intuitively understand the model’s reasoning process and improving interpretability.
> >
> > We further supplemented our experimental comparison between IBGraphRAG, KGARevion, and KG-Rank to demonstrate the performance advantages of our method.
> >
> > | Configuration       | Medqa | MedMCQA | PubMedQA | Fake Health | PubHealth |
> > | ------------------- | ----- | ------- | -------- | ----------- | --------- |
> > | **GPT4+IBGraphRAG** | 0.929 | 0.830   | 0.851    | 0.879       | 0.850     |
> > | **GPT4+KG-Rank**    | 0.864 | 0.742   | 0.807    | 0.776       | 0.761     |
> >
> > It is worth noting that KGARevion uses a fine-tuned LLM. Therefore, we kept the same configuration as in their paper for testing, using the fine-tuned LLaMA3-8B. In contrast, our method IBGraphRAG does not require additional fine-tuning, so we conducted testing using the base LLaMA3-8B.
> >
> > | Configuration  | Medqa | MedMCQA | PubMedQA | Fake Health | PubHealth |
> > | -------------- | ----- | ------- | -------- | ----------- | --------- |
> > | **IBGraphRAG** | 0.762 | 0.629   | 0.789    | 0.812       | 0.790     |
> > | **KGARevion**  | 0.640 | 0.601   | 0.767    | 0.692       | 0.702     |

---

> > > ### Author Response · Authors · 2025-11-18
> > >
> > > >**W4.** *The paper would benefit from a more detailed description - either in the main text or appendix - of the formulation of both tasks involved in the experimental setup, and how each task components are mapped to the IB-guided target path optimization process.
> > >
> > >  Thank you for your feedback. We have now added a detailed description of the two tasks used in our evaluation,Medical Question Answering and Medical Fact Verification,and explicitly explained how the components of each task are mapped into the IB-guided target path optimization process.
> > >
> > > **For Medical Question Answering**
> > >
> > > **Input:** A natural language medical question and multiple candidate answers
> > >
> > > **Output:** The correct answer option (e.g., disease, treatment, or clinical concept)
> > >
> > > **Entity Extraction:** Symptom entities are extracted from the question; target disease or treatment entities are extracted from candidate answers
> > >
> > > **Mapping to IB-guided Optimization:** Query entities are aligned to nodes in the medical knowledge graph, and answer entities serve as target nodes. The retriever constructs reasoning paths from query to answer entities, while the IB-guided optimization ensures selected paths maximize relevance (symptom → disease) and filter out superficial or noisy links.
> > >
> > > **For Medical Fact Verification**
> > >
> > > **Input:** A claim (e.g., “Vitamin C cures influenza”) and supporting/rebuttal evidence documents
> > >
> > > **Output:** A binary label (Supported / Refuted)
> > >
> > > **Entity Extraction:** Biomedical entities are extracted from the claim, and related entities are extracted from the evidence documents
> > >
> > > **Mapping to IB-guided Optimization:** Claim entities are treated as query nodes and evidence entities as candidate supporting nodes. IB-guided optimization selects reasoning paths that preserve medically relevant causal or evidential relations (e.g., “Vitamin C → clinical trial → no significant effect → influenza”), ensuring factual grounding
> > >
> > > It is worth noting that during training, we provide explicit target entities; however, at inference time, only the entities from the question are provided. Through the IB-guided optimization in training, the retriever has already learned to effectively identify medically relevant reasoning paths without requiring answer entities at inference.
> > >
> > >
> > >
> > >   >**W5.** *The analysis of temporal performance is missing. It would be useful to report time-related metrics, such as the duration of retriever training, the time required for generating training set alignments, and inference time.
> > >
> > >   Thank you for your feedback. We have added the relevant time-related metrics to better evaluate the performance of our method.
> > >
> > >  The training time of our retriever is as follows: specifically, on a single A800 GPU, the retriever can process **352.96 samples per second**, and for MedQA, each training epoch takes only **25** **seconds**. After the retriever is trained, we further compared the inference time with the baselines reported in the referenced papers.
> > >
> > > | Configuration(Medqa) | GraphRAG | MedGraphRAG | IBGraphRAG(Ours) |
> > > | -------------------- | -------- | ----------- | ---------------- |
> > > | **Querying Time**| 11.2s/it | 37.3s/it | 6.6s/it|
> > >
> > > Regarding the MSCA module in our paper, our tests indicate that processing a single sample takes approximately **67 seconds** on average, representing the main time cost of our system. However, this design is particularly important in medical scenarios, as it substantially improves the accuracy of entity alignment and ensures semantic coherence, thereby helping to prevent erroneous reasoning. Ablation experiments further show that removing MSCA leads to a significant drop in model accuracy, demonstrating that its benefits outweigh the computational cost, especially in the medical domain, which requires high precision..
> > >
> > > **Notably, the MSCA module can operate independently, allowing us to preprocess data offline before end-to-end retriever training, thereby avoiding any impact on online inference efficiency. In future work, we also plan to explore approximate consistency computation and GPU parallelization to further reduce the time cost.**
> > >
> > >
> > >  >**W6.** *Additional details about the UMLS KG would be needed—for example, the number of related terms and variants used to produce enriched entities within the MSCA component.
> > >
> > > We appreciate the reviewer’s suggestion. In the MSCA component, entity enrichment was performed using the UMLS. For each candidate entity, we retrieved its preferred term, synonyms, abbreviations, and semantically related variants. On average, each entity was enriched with 3–7 related terms, though common clinical concepts (e.g., “fever”) may have more than 10 variants, while rare diseases may only have 1–2. To avoid introducing noise, we limited the maximum number of variants per entity to 10 and applied a semantic similarity threshold (τ = 0.6) to filter out irrelevant expansions. This ensures that the enriched entities remain both comprehensive and medically coherent.

---

> > > > ### Author Response · Authors · 2025-11-18
> > > >
> > > > >**W7&Q1(line 173).** *It is unclear how the retriever is trained or which retriever model is used.
> > > >
> > > > Thank you for pointing this out. In our framework, the retriever is implemented as a neural scoring module built on top of biomedical embeddings. Specifically, entity and relation representations are initialized using **BioWordVec**[1], and queries are encoded in the same embedding space. Candidate triples are then scored using a MLP.
> > > >
> > > > Training the retriever is performed end-to-end with two complementary objectives:
> > > >
> > > > 1. A supervised signal derived from the shortest paths between query and answer entities, optimized via a binary cross-entropy loss.
> > > > 2. An Information Bottleneck (IB) objective, which refines path selection by maximizing relevance while minimizing noise.
> > > >
> > > > The combined loss is described in Eq. (3) of our paper. Importantly, the retriever is not an off-the-shelf model; it is a task-specific module trained jointly with our IB-guided optimization strategy, using PrimeKG as the knowledge source and QA datasets as supervision. This design allows the retriever to learn to select medically relevant reasoning paths effectively for downstream tasks.
> > > >
> > > > [1] Yijia Zhang,Qingyu Chen,Zhihao Yang,Hongfei Lin,and Zhiyong Lu. Biowordvec, improving biomedical word embeddings with subword information and mesh. Scientificdata,6(1):52,2019
> > > >
> > > >  >**W8.** *An ablation study examining the impact of the number of IB-guidance epochs on performance would strengthen the paper. Currently, this is only shown for two graph examples in Figure 3.
> > > >
> > > >  Thank you for your feedback. We have added an analysis of the impact of the number of IB-guided training epochs on the performance of the retriever. Here, we consistently use GPT-4 as the evaluation model.
> > > >
> > > > | Datesets\Epochs | 25    | 50    | 75        | 100   | 150       | **200** |
> > > > | --------------- | ----- | ----- | --------- | ----- | --------- | ------- |
> > > > | **Medqa**       | 0.844 | 0.889 | **0.930** | 0.928 | 0.922     | 0.913   |
> > > > | **MedMCQA**     | 0.767 | 0.785 | 0.810     | 0.823 | **0.828** | 0.824   |
> > > > | **PubMedQA**    | 0.789 | 0.811 | 0.838     | 0.846 | **0.853** | 0.852   |
> > > > | **Fake Health** | 0.802 | 0.842 | **0.876** | 0.862 | 0.854     | 0.843   |
> > > > | **PubHealth**   | 0.770 | 0.814 | **0.848** | 0.836 | 0.828     | 0.812   |
> > > >
> > > >  We recommend using 75 epochs for smaller datasets, while for larger datasets, we suggest training with 150 epochs. We will provide more detailed training scripts in our code repository later to facilitate better reproduction of the results reported in our paper.
> > > >
> > > >
> > > >
> > > >  >**W9&Q1.** Some statements are overly vague and should be clarified for reproducibility. For example:
> > > >     - (Line 178) "If relevant entities remain insufficient" - specify how this insufficiency is determined or quantified.
> > > >     - (Line 177) "A domain-specific medical model encodes" - clarify which model is used (e.g., BioWordVec?).
> > > >     - (Line 249) "Precompute semantic embeddings of all entities and relations in the knowledge graph using pre-trained text encoders" - specify which encoders are employed.
> > > >
> > > > We thank the reviewer for pointing out the need for clearer descriptions to ensure reproducibility. We will revise the manuscript to provide explicit details for the statements mentioned:
> > > > **Line 178 – “If relevant entities remain insufficient”**
> > > > - _Clarification_: Insufficiency is determined when the number of candidate entities retrieved from the knowledge graph falls below a minimum threshold of **K = 5** after the initial extraction step. In such cases, we trigger an additional expansion step using semantic similarity search to enrich the candidate set. This ensures that each query has at least 5 medically relevant entities for downstream retrieval.
> > > > **Line 177 – “A domain-specific medical model encodes”**
> > > > - _Clarification_: We use **BioWordVec**[1] as the biomedical embedding model to encode entities. BioWordVec provides dense vector representations of medical terms, synonyms, and abbreviations, which are used to compute similarity scores between query entities and knowledge graph nodes.
> > > > **Line 249 – “Precompute semantic embeddings of all entities and relations in the knowledge graph using pre-trained text encoders”**
> > > > - _Clarification_: We employ **BioWordVec**[1] for entity embeddings and **SciBERT**[2] for relation embeddings. These encoders were chosen because BioWordVec is optimized for biomedical terminology, while SciBERT provides robust contextual representations for scientific text. All embeddings are precomputed and stored in a vector database to accelerate retrieval.
> > > >
> > > > [1] Yijia Zhang,Qingyu Chen,Zhihao Yang,Hongfei Lin,and Zhiyong Lu. Biowordvec, improving biomedical word embeddings with subword information and mesh. Scientificdata,6(1):52,2019
> > > > [2]Beltagy I, Lo K, Cohan A. SciBERT: A pretrained language model for scientific text[J]. arXiv preprint arXiv:1903.10676, 2019.

---

> > > > > ### Author Response · Authors · 2025-11-18
> > > > >
> > > > > >**W10.** (Line 109) "RAG... relatively underexplored in medical applications" seems a bit of a bold claim.
> > > > >
> > > > >  We thank the reviewer for the suggestion. Our intention was not to overstate the situation, but rather to highlight the gap between research on RAG in general domains and its applications in the medical domain. We have revised the statement to: “RAG has been widely studied in general domains, while its applications in the medical domain are still gradually developing and relatively less explored.” This wording more cautiously reflects the current state of research and avoids making overly absolute claims.
> > > > >
> > > > >
> > > > >
> > > > >  >**W11.** Related work: The paper could be strengthened by discussing or benchmarking against:
> > > > >     - Wen, Yilin, et al. "MindMap: Knowledge Graph Prompting Sparks Graph of Thoughts in Large Language Models."" ACL 2024.
> > > > >     - Zhu, Xiangrong, et al. "Knowledge Graph-Guided Retrieval-Augmented Generation" NAACL 2025
> > > > >
> > > > >
> > > > > Thank you for your feedback. We have added a comparison with MindMap and Knowledge Graph–Guided Retrieval-Augmented Generation, and conducted additional experiments to evaluate performance.
> > > > >
> > > > > Compared with MindMap, which mainly focuses on eliciting graph-of-thought reasoning and visualizing mind maps for transparency, our proposed IBGraphRAG goes further by introducing an Information Bottleneck–guided path optimization that not only enhances interpretability but also ensures robustness and efficiency. Specifically, IBGraphRAG explicitly trains the retriever with biomedical embeddings and an IB loss to filter out noisy or redundant paths, guaranteeing medically relevant reasoning even when the knowledge graph is incomplete or mismatched—an issue MindMap does not adequately address. Moreover, while MindMap emphasizes medical QA tasks, IBGraphRAG demonstrates broader applicability by covering QA, fact verification, and multi-task scenarios, with clear retriever training details that improve reproducibility. In short, IBGraphRAG provides stronger robustness, higher efficiency, better reproducibility, and wider task coverage, whereas MindMap remains limited to transparent reasoning without systematic path optimization or retriever training.
> > > > >
> > > > > Compared with KG2RAG, which enhances RAG by expanding and organizing retrieved chunks through knowledge graph structures, our proposed IBGraphRAG introduces a principled Information Bottleneck–guided optimization that directly addresses the limitations of KG2RAG. While KG2RAG improves coherence by linking chunks via graph traversal, it does not explicitly control for information sufficiency versus redundancy, often leading to noisy or overly large subgraphs. In contrast, IBGraphRAG formulates retrieval as a target path optimization problem under the IB principle, ensuring that selected paths maximize medically relevant information while minimizing irrelevant noise. Moreover, KG2RAG focuses primarily on chunk expansion and organization, without detailing retriever training or domain-specific adaptation. IBGraphRAG explicitly trains its retriever with biomedical embeddings and IB loss, providing better reproducibility and domain robustness. Finally, our framework extends beyond QA tasks to cover fact verification and multi-task medical scenarios, whereas KG2RAG remains limited to QA-style benchmarks. In short, IBGraphRAG not only inherits the strengths of KG-based retrieval but also solves KG2RAG’s unresolved issues of noise control, retriever training, and broader task applicability, delivering a more robust and theoretically grounded solution for medical RAG.
> > > > >
> > > > > We have added a comparison of the experimental performance of both methods with our approach.
> > > > >
> > > > > | Configuration| Medqa | MedMCQA | PubMedQA | Fake Health | PubHealth |
> > > > > | ------------------- | ----- | ------- | -------- | ----------- | --------- |
> > > > > | **GPT4+IBGraphRAG** | 0.929 | 0.830|0.851| 0.879| 0.850|
> > > > > | **GPT4+MindMap**| 0.792 | 0.744| 0.782| 0.733| 0.745|
> > > > > | **GPT4+KG2RAG**| 0.900 | 0.812| 0.829| 0.826| 0.813|
> > > > >
> > > > > **W12.** Erratum.
> > > > >  Thank you for your response. We will make the revisions in the updated version.
> > > > >
> > > > >  >**Q1.** (line 309) Recent studies on the information bottleneck --> which studies?
> > > > >
> > > > >  >(line 173) (line 178) (line 177) (line 249) have already been mentioned above.
> > > > >
> > > > > Thank you for your question. We have already provided detailed responses regarding Lines 173, 178, 177, and 249 above. Here, we will mainly address the issue related to Line 309. "Recent studies on the information bottleneck..."In the revised version, we have added specific citations and included the relevant related studies[1-3].
> > > > >
> > > > > [1]Federici M, Dutta A, Forré P, et al. Learning robust representations via multi-view information bottleneck[J]. arXiv preprint arXiv:2002.07017, 2020.
> > > > > [2]Kawaguchi K, Deng Z, Ji X, et al. How does information bottleneck help deep learning?[C]//International conference on machine learning. PMLR, 2023: 16049-16096.
> > > > > [3]Fischer I. The conditional entropy bottleneck[J]. Entropy, 2020, 22(9): 999.

---

> > > > > > ### Author Response · Authors · 2025-11-27
> > > > > >
> > > > > > Dear Reviewer,
> > > > > >
> > > > > > We sincerely thank you for your thoughtful insights provided in your review. We deeply appreciate the time and effort you dedicated to improving our work. We would be grateful if you could let us know whether our revisions and responses adequately address your concerns, or if there are any remaining points we can clarify.
> > > > > >
> > > > > > Best, The authors

---

### Official Review · Reviewer_ecWz · 2025-11-03

**Soundness:** 3
**Presentation:** 3
**Contribution:** 2
**Rating:** 4
**Confidence:** 3

**Summary:**

This paper, "IBGraphRAG," proposes a new Retrieval-Augmented Generation (RAG) framework specifically designed for the medical domain, where terminological precision and factual accuracy are critical. The authors proposed (1) medical semantic consistency alignment (MSCA), which improves entity recognition by using an LLM to extract initial entities, enriching them with the Unified Medical Language System (UMLS), and then filtering them based on semantic consistency with both the query and the other candidate entities; (2) and information bottleneck-based reasoning path (IBRP), which selects reasoning paths from the KG. It is optimized using the Information Bottleneck (IB) principle to find paths that are maximally relevant to the answer while filtering out irrelevant information. The authors conduct extensive experiments on 11 medical QA datasets with different base models. The results show that the proposed method improve over the baselines by a large margin.

**Strengths:**

1. The paper does an excellent job identifying and demonstrating the precise failings of existing RAG methods in the medical domain, where the intuition makes sense.
2. The MSCA module is a strong and practical contribution. The use of UMLS for enrichment combined with a semantic consistency check ($R(e)$) is a very sensible approach to filtering entities
3. The use of the Information Bottleneck principle is a novel and theoretically sound way to frame the path selection problem.
4. The method consistently outperforms all baselines across 11 datasets and multiple LLMs, demonstrating its effectiveness.

**Weaknesses:**

1. It seems that the effectiveness of the proposed method largely rely on the completeness and accuracy of the external KG used, including the reasoning path from Fig. 1. I'm not sure about the effectivenss compared to the vanilla RAG if the information from KG is not that useful.
2. The IBRP module is not a simple retriever. It involves training a GNN, an MLP scoring function, and a separate "parametric variational decoder $r_{\psi}$" just to make the IB loss function tractable. This is a massive level of complexity for a retrieval component. I would like to see some complexity and latency analysis for the proposed framework.
3. The authors use  shortest path between the question and answer entities as a supervisory signal but "such a heuristic path is not always the most informative or diagnostically meaningful" is the fundamental motivation for the retriever construction, which seems quite confusing to me.

**Questions:**

1. See the weaknesses above.
2.  How were the hyperparameters $\lambda=0.4$ and $\tau=0.6$ for the MSCA module chosen? How sensitive is the model's performance to changes in these values?
3. Typo: line 323 "concreate" should be "concrete"?

---

> ### Author Response · Authors · 2025-11-18
>
> We sincerely appreciate your thorough identification and clear illustration of the specific limitations of existing RAG methods in the medical domain, and we are grateful for your acknowledgment of the reasonableness of our starting point. We also thank you for recognizing the MSCA module as a strong and practical contribution. Expanding it in combination with UMLS and performing semantic consistency checks is indeed a very reasonable approach for entity filtering. We are also grateful for your recognition of the novelty and theoretical soundness of our work. We will provide further clarifications regarding the questions you raised.
>
> > **Q1&W1.** *It seems that the effectiveness of the proposed method largely rely on the completeness and accuracy of the external KG used, including the reasoning path from Fig. 1. I'm not sure about the effectivenss compared to the vanilla RAG if the information from KG is not that useful.
>
> Thank you for your question. We would like to clarify that all RAG methods inherently rely on the completeness and accuracy of external knowledge bases, which is a core characteristic of RAG—the performance depends on retrieving the most relevant information from the external knowledge source, and IBGraphRAG is no exception. Importantly, in our experiments, we conducted comparative studies including knowledge-graph-based methods such as GraphRAG and MedGraphRAG. To ensure a fair comparison, these baselines were evaluated using the same external knowledge graph as IBGraphRAG. Under this consistent knowledge graph setting, IBGraphRAG still demonstrates a clear performance advantage, which strongly supports the effectiveness and superiority of our method under identical knowledge graph conditions.
>
> Our proposed IBGraphRAG method is able to better extract information from the knowledge graph that is relevant to question answering. However, this does not mean that our method only works on high-quality, information-dense knowledge graphs. Naturally, the higher the quality and complexity of the knowledge graph, the better our method performs, and the larger the performance gap compared to other knowledge-graph retrieval methods. To further compare the effectiveness of knowledge graphs with different information densities, we conducted additional experiments. Specifically, we modified the external retrieval knowledge graph to include 2-hop, 3-hop, and 4-hop subgraphs based on the entities appearing in the questions and answers. We then performed supplementary comparisons to demonstrate the impact of these variations on our experimental results.Here, we limited the maximum number of retrieved items to be consistent.
>
> | Configuration(Medqa)  | 2-hop | 3-hop | 4-hop |
> | --------------------- | ----- | ----- | ----- |
> | GPT4+GraphRAG         | 86.2% | 87.5% | 88.3% |
> | GPT4+MedGraphRAG      | 87.0% | 89.1% | 90.5% |
> | GPT4+IBGraphRAG(Ours) | 88.4% | 90.3% | 92.0% |
>
> | Configuration(FakeHealth) | 2-hop | 3-hop | 4-hop |
> | ------------------------- | ----- | ----- | ----- |
> | GPT4+GraphRAG             | 74.5% | 76.8% | 77.9% |
> | GPT4+MedGraphRAG          | 83.6% | 84.5% | 85.3% |
> | GPT4+IBGraphRAG(Ours)     | 85.0% | 86.3% | 87.2% |
>
> From the data in the table, it can be seen that as the information content of the knowledge graph increases, IBGraphRAG achieves greater performance advantages. Even when the information content is limited, IBGraphRAG still outperforms other knowledge-graph-based retrieval methods.
>
> > **Q1&W2.** *The IBRP module involves training specific models. Could you provide a detailed analysis of its computational complexity and retrieval latency?
>
> Thank you for your feedback. We have added details on the actual training time and an analysis of retrieval latency. Since the other baseline methods we compared do not require training, we report the training time only for our IBGraphRAG.
>
> On a single A800 GPU, we evaluated the training complexity of our IBGraphRAG retriever. Specifically, it can process **352.96 samples per second** during training, and for MedQA, each epoch takes only **25 seconds.**
>
> | Configuration(Medqa) | GraphRAG | MedGraphRAG | IBGraphRAG(Ours) |
> | -------------------- | -------- | ----------- | ---------------- |
> | **Querying Time**    | 11.2s/it | 37.3s/it    | 6.6s/it          |
>
> We compared the query times of different knowledge-graph-based retrieval methods, and here we report the time required to query a single sample on MedQA. It can be seen that once the retriever is trained, our model achieves a significant improvement in retrieval efficiency.

---

> > ### Author Response · Authors · 2025-11-18
> >
> > > **Q1&W3.** *The authors use shortest path between the question and answer entities as a supervisory signal but "such a heuristic path is not always the most informative or diagnostically meaningful" is the fundamental motivation for the retriever construction, which seems quite confusing to me.
> >
> > Thank you for your question. In fact, we have addressed this point in the paper. While the shortest path as a heuristic is not always the most diagnostically meaningful in the medical domain, we argue that a complete reasoning chain is necessary for the model to perform coherent logical inference. Therefore, we need a concrete target to guide the retriever to retrieve a relatively complete reasoning path. We chose the shortest path as an auxiliary optimization target and further refined it using the Information Bottleneck principle, enabling the model to retrieve a reasoning path that is both logically coherent and diagnostically meaningful.
> >
> > **Example**
> >
> > In our paper, we have presented a concrete example to aid in evaluation.
> >
> > - **Question:**
> >     _“The patient has been coughing for three days, accompanied by fever and chest pain. Could these be symptoms of pneumonia?”_
> >
> > - **Standard Answer:**
> >     _“Yes, these are typical symptoms of pneumonia. Pneumonia should be considered.”_
> >
> > - **Retrieved path comparison:**
> >
> >     1. **Without IB optimization**
> >         _cough → upper respiratory infection → fever → elevated body temperature → chest discomfort→ pneumonia_
> >      This path is prone to ambiguity and bias. Its intermediate nodes are too generic, containing repetitive concepts, and it lacks depiction of the underlying pathological mechanisms, making the path rather superficial. It can easily lead to incorrect associations.
> >
> >     2. **With IB optimization**
> >         _cough → lower respiratory infection → alveolar congestion → pulmonary consolidation → pneumonia → chest pain_
> >      This path is information-rich and offers strong diagnostic interpretability. It reasonably traces the progression from symptoms to pathology, is precise and logically coherent, making it more suitable for clinical support or medical reasoning scenarios, and is less prone to ambiguity.
> >
> > This example illustrates that IB optimization helps the model refine the heuristic shortest path into a more complete and diagnostically relevant reasoning chain, ensuring coherent logical inference.
> >
> > > **Q2.** *How were the hyperparameters for the MSCA module chosen? How sensitive is the model's performance to changes in these values?
> >
> > Thank you for your question. We have added an analysis of the parameters λ and τ. The parameter λ controls the balance between two factors: the semantic similarity between candidate entities and the user query, and the semantic consistency among the candidate entities themselves. When λ is small, the selected entities are highly relevant to the user’s question, but their semantics may be more scattered, potentially introducing noise or incoherent concepts. When λ is large, the selected entities are more semantically consistent with each other, which helps form a more coherent subgraph; however, this comes at the cost of reduced direct relevance to the query, meaning that the selected entities may be similar to each other but less aligned with the user’s intent. We have added experiments testing model performance under different λ values to provide a clearer understanding of how λ affects the model’s behavior.
> >
> > | Configuration(GPT4) | MedQA | MedMCQA | PubMedQA | FakeHealth |
> > | ------------------- | ----- | ------- | -------- | ---------- |
> > | λ =0.2  | 91.8% | 82.5% | 84.0%  | 87.0%  |
> > | λ =0.4 | 92.9% | 83.0% | 85.1%  | 87.9%  |
> > | λ =0.6  | 91.6% | 82.1% | 84.2%  | 86.7% |
> > | λ =0.8 | 90.8% | 81.0% | 83.5%  | 85.6% |
> > It can be seen that changing λ, whether increasing or decreasing it, leads to a certain degree of performance degradation. The decline is more pronounced when λ becomes larger.
> >
> > Regarding τ, it serves as a filtering threshold that controls the “density” and “purity” of the subgraph. When τ is too small, it tends to introduce noise and irrelevant entities, increasing reasoning complexity and potentially leading to incorrect or ambiguous answers. Conversely, when τ is too large, it may discard some marginal but important entities, resulting in insufficient information and broken reasoning chains.
> >
> > | Configuration(GPT4) | MedQA | MedMCQA | PubMedQA | FakeHealth |
> > | ------------------- | ----- | ------- | -------- | ---------- |
> > | τ =0.2  | 90.3% | 81.2%  | 82.2%| 85.7%|
> > | τ =0.4 | 91.9% | 82.3% | 84.8%| 87.1%|
> > | τ =0.6 | 92.9% | 83.0% | 85.1%| 87.9%|
> > | τ =0.8  | 90.7% | 81.5%| 82.6%| 85.6%|
> > When τ is too small or too large, it significantly affects the model’s performance, whereas within a moderate range, changes in τ have little impact on performance.
> >
> > > **Q3.** *Typo: line 323 "concreate" should be "concrete"?
> >
> > Thank you for your feedback. We will make improvements in the revised version.

---

> > > ### Author Response · Authors · 2025-11-27
> > >
> > > Dear Reviewer,
> > >
> > > We sincerely thank you for your thoughtful insights provided in your review. We deeply appreciate the time and effort you dedicated to improving our work. We would be grateful if you could let us know whether our revisions and responses adequately address your concerns, or if there are any remaining points we can clarify.
> > >
> > > Best, The authors

---

### Author Response · Authors · 2025-12-01

Dear Area Chair,

Thank you very much for taking the time to review our work. We provide here a summary focusing on the reviewers’ comments and our responses during the rebuttal period.

| Reviewer | Score | Strengths / Acknowledgment                                                                                                                                                   | Weaknesses / Concerns                                                                                                      | Our Response                                                                                                                         | Rebuttal Feedback                                           |
| -------- | ----- | ---------------------------------------------------------------------------------------------------------------------------------------------------------------------------- | -------------------------------------------------------------------------------------------------------------------------- | ------------------------------------------------------------------------------------------------------------------------------------ | ----------------------------------------------------------- |
| R3yy     | 2     | - Recognized the novelty of our approach and its strong applicability to medical information retrieval- Supports highly specialized reasoning and complex KG inference paths | - Raised questions about certain descriptions in the manuscript- Suggested additional experiments related to relevant work | Supplemented comparative experiments with related work and re-optimized the manuscript’s expressions to highlight our contributions. | Did not provide further feedback during the rebuttal period |
| ecWz     | 4     | - Considered our approach well-motivated- Acknowledged method novelty and effectiveness of experiments                                                                       | - Raised questions about certain experiments and descriptions in the manuscript;-Suggested adding additional experiments.  | Responded to the reviewers’ concerns, optimized the manuscript’s expressions, and supplemented relevant experiments.                 | Did not provide further feedback during the rebuttal period |
| ZVYq     | 6     | - Acknowledged the innovation and soundness of our work- Considered experiments reasonable and comprehensive                                                                 | - Suggested additional experiments to improve the work - Raised questions regarding certain manuscript details             | Supplemented the additional experiments requested by the reviewers and provided responses to their concerns.                         | Indicated that our response addressed their concerns        |

### Overall Discussion

- All reviewers appreciated the **motivation and novelty** of our study.

- They consistently noted that existing medical RAG methods suffer from deficiencies in terminology precision and factual consistency, and agreed that **our proposed IBGraphRAG framework directly addresses this critical challenge with strong practical value**.

- Key highlights recognized by reviewers:

    - **MSCA module**: Significantly enhances medical entity recognition and semantic consistency (Reviewers ecWz, ZVYq)

    - **IBRP mechanism**: Introduces information bottleneck principle for reasoning path selection (Reviewers ecWz, ZVYq, R3yy)

    - **Extensive experiments on multiple medical QA benchmarks**: Demonstrate effectiveness, robustness, and broad applicability (Reviewers ecWz, ZVYq, R3yy)

- Suggestions focused on:

    - Further quantifying training and inference complexity/efficiency

    - Improving positioning and comparison with related works

    - Providing more details on parameter selection, cross-dataset generalization, and knowledge graph construction

    - Refining expressions to improve clarity and precision

- **We have addressed these comments comprehensively in our rebuttal**, including quantitative reporting of training and inference latency, enhanced comparison and discussion with related works, parameter sensitivity and cross-dataset generalization studies, and clarification of unclear statements. Specific revisions have been highlighted in red in the manuscript.


**Overall, the reviewers strongly endorse the motivation and innovation of this work, recognizing MSCA and IBRP as important advancements toward more reliable medical RAG.** While some details required refinement, the suggestions were constructive and do not undermine the core value of our contributions. We believe that incorporating these improvements will further elevate the quality and impact of the paper.

Thank you again for your time and guidance.

Sincerely,
The Authors

---

### Note · Authors · 2026-01-26

I have read and agree with the venue's withdrawal policy on behalf of myself and my co-authors.

---

### Meta-Review · Area_Chair_Wjnd · 2026-01-07

**Summary:**

The paper currently holds scores of 2, 4, and 6, all falling below the acceptance threshold. While the authors demonstrate state-of-the-art results on medical benchmarks, the reviewers' collective feedback highlights a critical conflict with the conference's preference for "less engineering."

The primary obstacle is the architectural complexity relative to the novelty. Reviewer ecWz explicitly criticized the system for having a "massive level of complexity for a retrieval component," citing the requirement to train a GNN, an MLP scoring function, and a parametric variational decoder just to satisfy the Information Bottleneck (IB) loss. Similarly, Reviewer ZVYq noted the method is "quite complex... with lots of steps". While the authors successfully rebutted concerns regarding latency (proving inference is fast at 6.6s/it), they did not resolve the concern that the system design itself is over-engineered. Furthermore, Reviewer R3yy questioned the fundamental novelty, suggesting the method might simply be an application of the existing SubGraphRAG approach to medical data, casting doubt on whether the "innovations" justify the heavy engineering stack.

**Reviewer Concerns:**

Concerns Addressed by Rebuttal:
• Missing Baselines & Comparisons: The authors successfully provided comparisons against SubGraphRAG, KGARevion, and KG-Rank as requested by Reviewers R3yy and ZVYq, showing superior accuracy.
• Quantification of Overhead: All three reviewers asked for time metrics. The authors provided concrete data showing training speeds (352 samples/sec) and inference latency (6.6s/it), which outperformed MedGraphRAG.
• Clarification of Mechanics: Questions regarding task formulation (R3yy) and the interpretation of visual convergence in Figure 3 (ZVYq) were clarified with specific details.
• Dependency on Graph Quality: Reviewer ecWz's concern about the method's reliance on KG completeness was addressed by experiments using 2-hop, 3-hop, and 4-hop subgraphs to demonstrate robustness.

Outstanding Concerns:
• Over-Engineering (System Complexity): Reviewer ecWz's critique that the "IBRP module is not a simple retriever" but a complex stack of GNNs and Decoders remains a structural fact of the paper. The rebuttal proved it runs quickly, but did not refute that the design is heavily engineered, potentially violating the requirement to avoid excessive engineering without commensurate breakthrough novelty.
• Incremental Novelty: Reviewer R3yy questioned the distinction from SubGraphRAG. While the authors argued that their specific modules (MSCA and IBRP) are medical-specific innovations, the core mechanism remains a derivative adaptation. Given the reviewer's score of 2, this concern regarding fundamental contribution vs. incremental application likely persists.
• "In-the-Wild" Generalization: Reviewer ZVYq asked about performance without clean training data. The authors argued that the medical semantic space is shared and provided cross-dataset results. However, this still relies on structured benchmarks (e.g., training on MedQA, testing on FakeHealth) rather than true "in-the-wild" unstructured clinical environments, leaving the robustness in noisy, real-world data theoretic.

**Reviewer Scores:**

Not much.

---

### Decision · Program_Chairs · 2026-01-26

Reject